# Unchanged frequency and decreasing magnitude of outbursts from ice-dammed lakes in Alaska

B. Rick [1,2] ✉, D. McGrath [1], S. W. McCoy [3] & W. H. Armstrong[4]

Glacial lakes can form and grow due to glacial retreat, and rapid lake drainage can produce destructive floods. Outburst flood compilations show a temporal increase in frequency; however, recent studies highlight the role of observational bias, creating uncertainty about current and future glacial-lake hazards. Here, we focus on the Alaska region, which generated a third of previously documented outbursts globally. Using multitemporal satellite imagery, we documented 1150 drainages from 106 ice-dammed lakes between 1985 and 2020. Documented events became more frequent over time, however, accounting for increasing image availability reveals no significant increase occurred. Most lakes decreased in area and volume, suggesting a reduction in regional flood hazard. Our satellite-based approach documented 60% more events in a 35-year period than had previously been documented over 100 years. This suggests that outburst floods have historically been underreported and warrants systematic study of other regions.

Glacial lake outburst floods (GLOFs) occur when a lake dam fails or is overtopped, releasing a large volume of water to downstream environments[1]. These events can be triggered by mass movements into a lake (i.e., avalanche, landslide, rock fall), earthquakes, glacier calving, extreme rainfall or snowmelt, flotation of an ice dam, or degradation of an ice-cored moraine[2]. GLOFs can cause devastating impacts to infrastructure (e.g., roads, bridges, hydropower systems), ecosystems (i.e., altering sediment flux, salmon habitat), and livelihoods (i.e., fishing, agriculture)[3–8]. The regional GLOF hazard is a combination of both the frequency and magnitude of these events. Glacial lakes have been rapidly increasing in number (53%) and total area (51%) globally between 1990 and 2018[9]. This increase has led to predictions that with continued climate warming and glacier mass loss the frequency and magnitude of GLOF events and thus associated hazards will also increase[9–12]. Indeed, a recent global compilation of historical GLOFs shows that frequency of reported events has increased from ~1900 to 2017[13]. However, much of the observed increase in frequency could be explained by an increase in the number of scientific studies, leaving actual trends of frequency uncertain[13].

The most common dam types of flood-originating glacial lakes are moraines and glacial ice. Dam type influences the nature of the flood, its likelihood for repeat events[4,13], and is linked to rates of long-term lake area change[14]. Ice-dammed lakes are impounded behind a glacier dam which can form due to tributary valley deglaciation, a surging glacier blocking a drainage channel, or a small pocket forming along a valley wall[11]. Drainage results from flotation of the ice dam, the opening of subglacial conduits, complete ice dam failure, or flow over or along the ice margin[15–17]. Because subsequent ice flow can effectively close en- or subglacial tunnels and reseal the ice-dam, ice-dammed lakes often experience multiple fill and drain cycles. Drainages from ice-dammed lakes most commonly occur every 1–3 years[18], though some lakes can experience multiple events per year (e.g., Suicide Basin[19] in Alaska, Lake No Lake[20] in British Columbia, Lake Kyagar and Lake Merzbacher in High Mountain Asia[21]). In contrast, moraine-dammed lakes typically drain only once as permanent dam failure is common in the drainage process.

The timing of when a lake releases water can influence downstream impacts, depending on the conditions of the stream or river

[1]Department of Geosciences, Colorado State University, Fort Collins, CO 80523, USA. [2]Alaska Climate Adaptation Science Center, Fairbanks, AK 99775, USA. [3]Department of Geological Sciences and Engineering, University of Nevada, Reno, NV 89557, USA. [4]Department of Geological and Environmental Sciences, Appalachian State University, Boone, NC 28607, USA. ✉e-mail: bri.rick07@gmail.com

into which it drains. If river ice is still present, the additional water may cause enhanced flooding, whereas if the river stage is low the lake drainage may have little flooding impact[15]. The timing can also exacerbate ecological impacts; for example, streambed scour can disturb salmon eggs within streambed gravels and decrease wild salmon productivity[22,23].

In Alaska, moraine-dammed is the most common type of contemporary ice-marginal lake, composing over 40% of the lake population by number and nearly 80% of the total lake area[24]. However, since the 1980s, only four moraine-dammed lake drainage events have been documented[25]. In contrast, ice-dammed lakes compose 9% of the lake population and 6% of the total lake area in the 2010s[24], yet 339 drainage events have previously been documented from these lakes between 1985 and 2020[7]. Thus, ice-dammed lakes have caused a disproportionately high number of GLOF events in Alaska.

Rapid glacier retreat and thinning across Alaska[26,27] have contributed to the disappearance of ~50% of a representative subset of Alaskan ice-dammed lakes between the 1960s and 2008[14,24]. Although ice-dammed lakes in Alaska have decreased in both number and area since the 1980s[14], there has been a documented increase in the number of reported GLOF events[13], suggesting that more ice-dammed drainage events are produced by fewer lakes. Whereas there are Alaska-wide inventories of ice-dammed lakes that provide a basis for analyzing changes in lake size and location through time[14,15], the dynamics of ice-dammed lake drainages have not seen similar systematic study[25], likely due to a lack of widespread data availability with sufficient spatial and temporal resolution. Current understanding regarding the number, frequency, and characteristic initiation date of lake drainage events comes primarily from case studies of individual ice-dammed lakes[19,28–30] and from compilations of opportunistic observations with reporting bias towards events with societal impacts or lakes in more accessible locations[13,31]. This lack of systematic and spatially explicit observation of lake drainage events hinders the development of robust measures of lake drainage frequency and timing, critical inputs for accurate hazard assessments.

Focusing solely on ice-dammed lakes, here we document all identifiable lake drainage events over 1985–2020 in the Alaska region (including a small portion of NW Canada) using timelapse videos of satellite imagery (Landsat and Sentinel-2) produced in Google Earth Engine (see Methods). We manually analyze images to quantify drainage event frequency, timing, and size. We estimate lake volume and potential peak discharge from observed lake area for three different time periods (1950s, 1980s, and 2010s) using empirical scaling relationships. This study is the first to systematically and comprehensively document ice-dammed lake drainage events in the Alaska region and provides an assessment for how these events have changed over time, as well as a baseline and framework for future regional studies.

## Results and discussion
### Increase in detected drainage events over time attributable to imagery bias
We documented 1150 drainage events from 106 individual ice-dammed lakes across the Alaska region between 1985 and 2020. This is approximately three times the 339 events previously documented in the region over the same time period[13], though is likely still an underestimate. For the 20 lakes also monitored by the National Weather Service Alaska-Pacific River Forecast Center[18], 282 events were reported between 1985 and 2020. We observed 200 events for these 20 lakes via satellite imagery analysis, suggesting an underestimation of up to 30% (Fig. S1).

We documented an increase in the number of detected drainage events over time (Fig. 1b). Coincident with this increase, there have been increases in satellite image availability, mean annual air temperature in the region, and an increased rate of glacier mass loss. We find a strong positive correlation ($R^2 = 0.95$; Fig. S2) between the

number of lakes with adequate imagery available (see Methods) and the number of events detected. Mean annual air temperature also covaries with imagery and frequency of events over time, though it was found to be a weaker predictor ($R^2 = 0.64$) of event frequency than imagery availability ($R^2 = 0.77$). We find no clear relationship between event frequency and lake area, ice thinning rate, or ice thickness. These findings suggest that increased image availability is the primary driver of the increase in number of detected events with time. Image availability increased markedly with the launches of Landsat 7, Landsat 8, and Sentinel-2 in 1999, 2013, and 2015, respectively (Fig. 1c), with an average of 133 images over the study area per year between 1985 and 2000 and an average of 575 images per year between 2001 and 2020.

### Unchanged frequency of lake drainage events at the regional scale
To correct for the bias stemming from image availability, we assessed the change in frequency of events over time using two independent methods (see Methods). Method one looked at how many lakes had documented drainage relative to the number of lakes that had adequate imagery each year, while method two explored whether a trend existed over time if image availability remained constant (i.e., 50 images per year). Using method one, we find no statistically significant temporal trend in the proportion of lakes (ratio of number of events detected to number of lakes with adequate imagery) that drain each year (Fig. 1d). Best-fit linear trends in time have slopes approaching zero ($0.0018 \pm 0.0018$ events per year with $p$-value = 0.32. See Methods and Supplementary Figs. S3, S4 for details). Method two used a bootstrap method to remove bias stemming from image availability (see Methods), and found no increasing trend in event frequency over time ($-0.006 \pm 0.001$ events per year; Fig. S5). Collectively, these methods suggest that although glaciers have rapidly thinned and retreated in Alaska[27] and ice-dammed lakes have decreased in both number and area[14], the regional frequency of ice-dammed lake drainages each year has remained unchanged over the past 35 years. When averaged across the Alaska region, it is expected that ~50% of ice-dammed lakes drain in any given year and that the average recurrence interval of a drainage event for an individual lake is ~2 years.

### Earlier timing of individual lake drainage events
Although the regional frequency of lake drainage events per year is stationary across the record, the frequency and seasonal timing of individual lakes shows substantial variability (Figs. 1, 2, S8, S9). Of the lakes that experienced five or more events and had a statistically significant trend over time, 75% experienced earlier release dates. The trend in progressively earlier draining has been recorded at a global scale as well[4,31]. Release date is thought to be influenced by when the lake fills, lake water temperature, dam strength, and ice dynamics[32]. As ice dams thin and weaken, it requires less hydrostatic pressure to lift the ice and produce a drainage channel, suggesting that drainage events should occur earlier in the season if water inputs to the lake are constant in time. Additionally, earlier onset of snow[33] and ice melt likely leads to earlier water input and lake filling, contributing to earlier drainage dates as well.

While we do not identify trends in event frequency when averaged across the study area, some individual lakes showed an increase in drainage event frequency with time (Fig. S8). Summit Lake is an example of a lake which has been monitored since its first outburst flood in 1961[34–36]. Nearly annual drainages began in 1961, with progressively earlier and smaller flood events marked by a decrease in volume over time[36]. Summit Lake has drained annually since 1970 but has lost over 4 km² (~80%) in surface area since then[15]. Strandline Lake also exhibits increased drainage frequency, draining every 2–3 years between 2000 and 2010, but every year between 2013 and 2020 (Fig. 3). However, the volume of water released from Strandline Lake has decreased over time, likely resulting in decreased peak discharges.

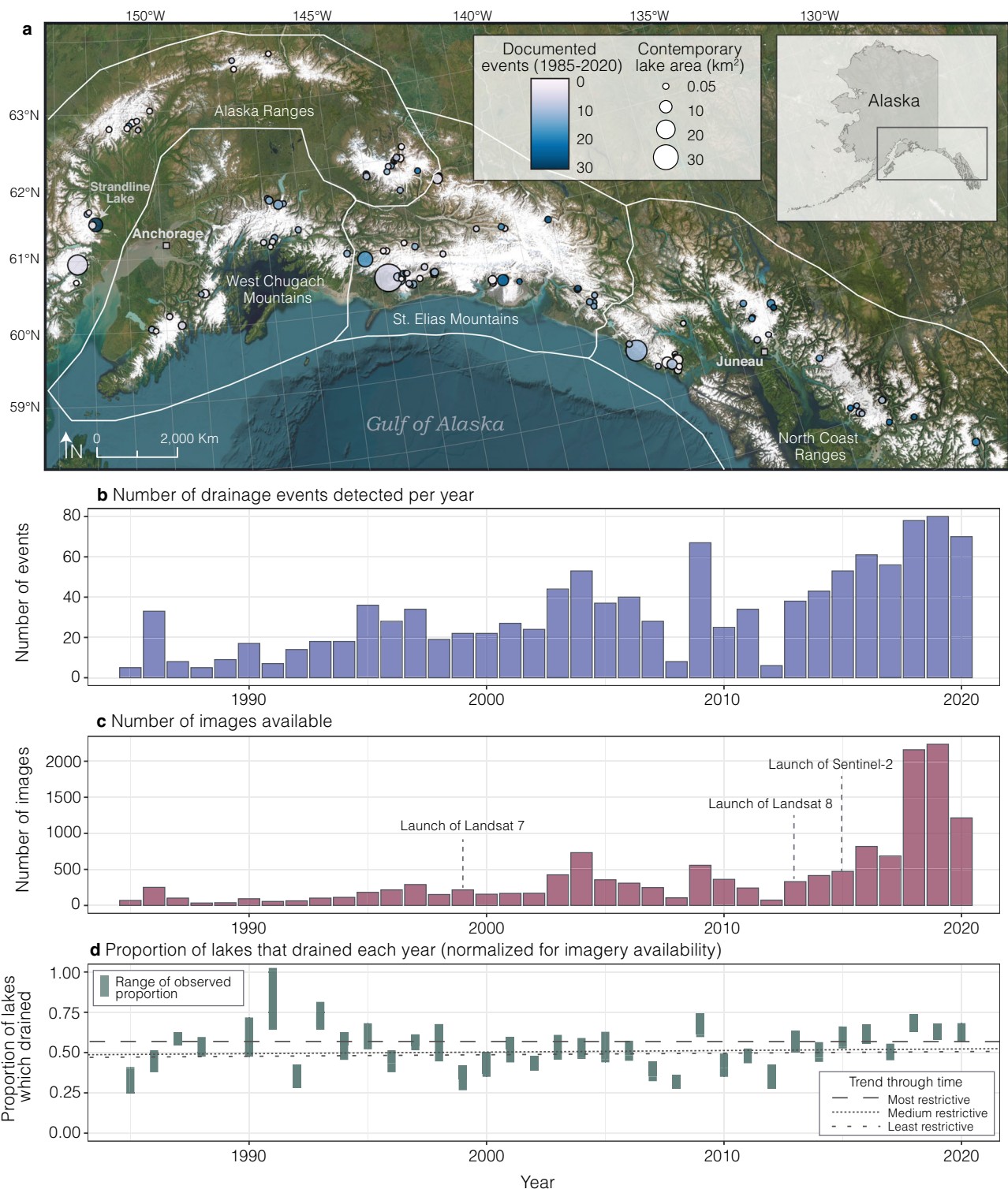

**Fig. 1 | Ice-dammed lake locations and drainage events through time. a** Spatial distribution of ice-dammed lakes colored by total number of events documented between 1985 and 2020, sized by area. **b** Total number of detected drainage events per year. **c** Total number of cloud-free images available per year. **d** Range of proportion of lakes that drained each year calculated as the ratio of number of events detected to number of lakes with adequate imagery given three different criteria (see Methods or Fig. S3). ArcGIS software by Esri was used to create the map. Basemap[55] imagery sources: Esri, Maxar, Earthstar Geographics, and the GIS User Community.

These examples illustrate the complexity of determining GLOF hazards, as both event frequency and magnitude combine to determine the hazard.

As glaciers continue to thin and retreat, lakes that fill and drain would be expected to move upwards in elevation along the glaciers[24], as glacier network structure in Alaska should allow for the creation of new lakes as tributary valleys deglaciate. The lifespan of a lake depends on the height and thinning rate of the ice dam, as a dam's height limits its ability to impound water[37]. An interesting question for future work is whether accelerating rates of ice loss in this

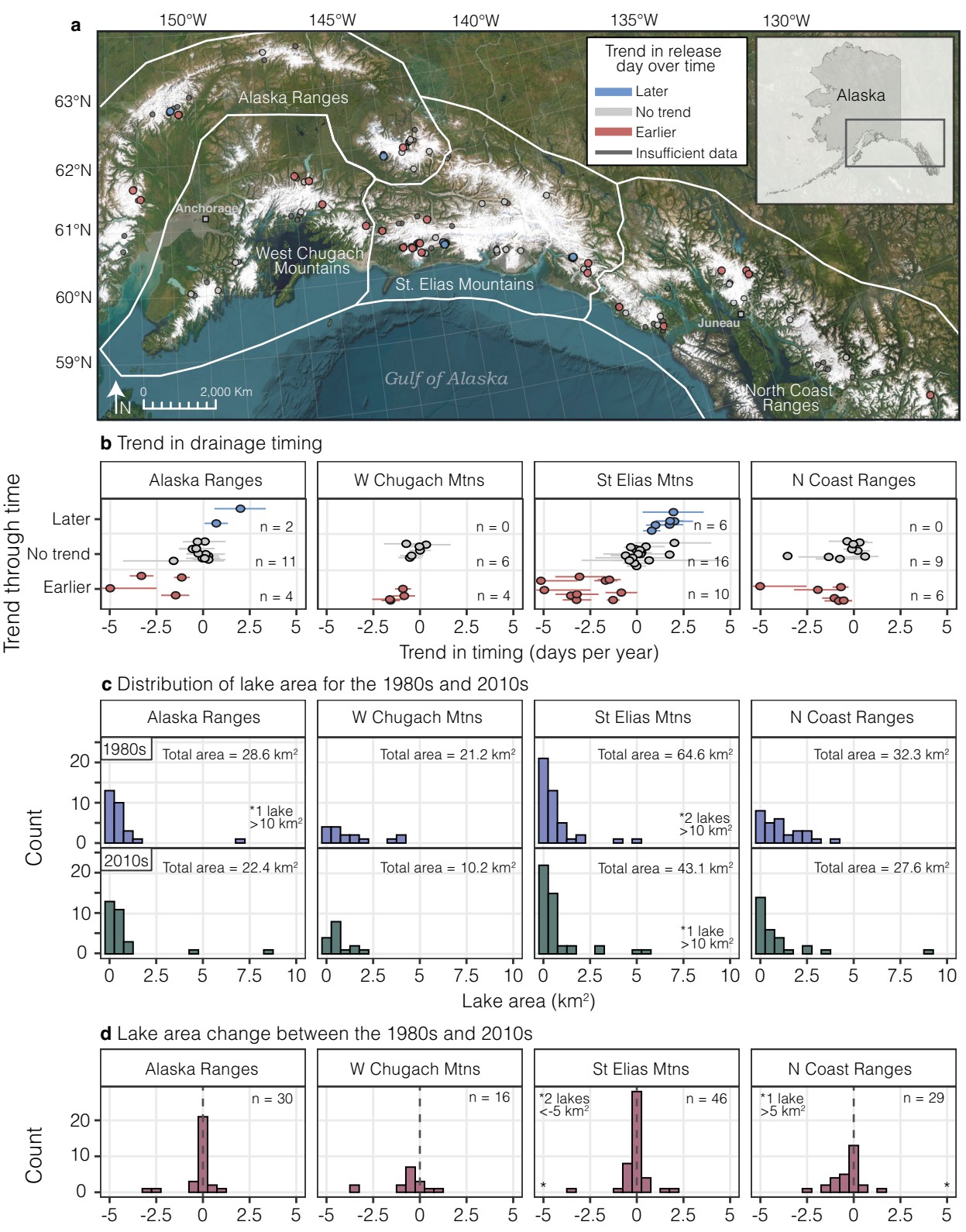

**Fig. 2 | Trend in lake release date and lake area over time. a** Spatial distribution of lakes colored by trend in drainage event timing (red = earlier release, gray = no significant trend, and blue = later release). **b** Trend in drainage event timing (in days per year) for each lake which experienced five or more drainage events, colored by whether the trend indicates an earlier (red), later (blue), or undeterminable (gray) release over time. Error bars indicate the ninety-five percent confidence interval.

See Fig. S9 for individual lake trends. **c** Histograms showing the distribution of lake areas for the 1980s and 2010s for each subregion. **d** Histograms showing the distribution of lake area change between the 1980s and 2010s for each subregion. ArcGIS software by Esri was used to create the map. Basemap[55] imagery sources: Esri, Maxar, Earthstar Geographics, and the GIS User Community.

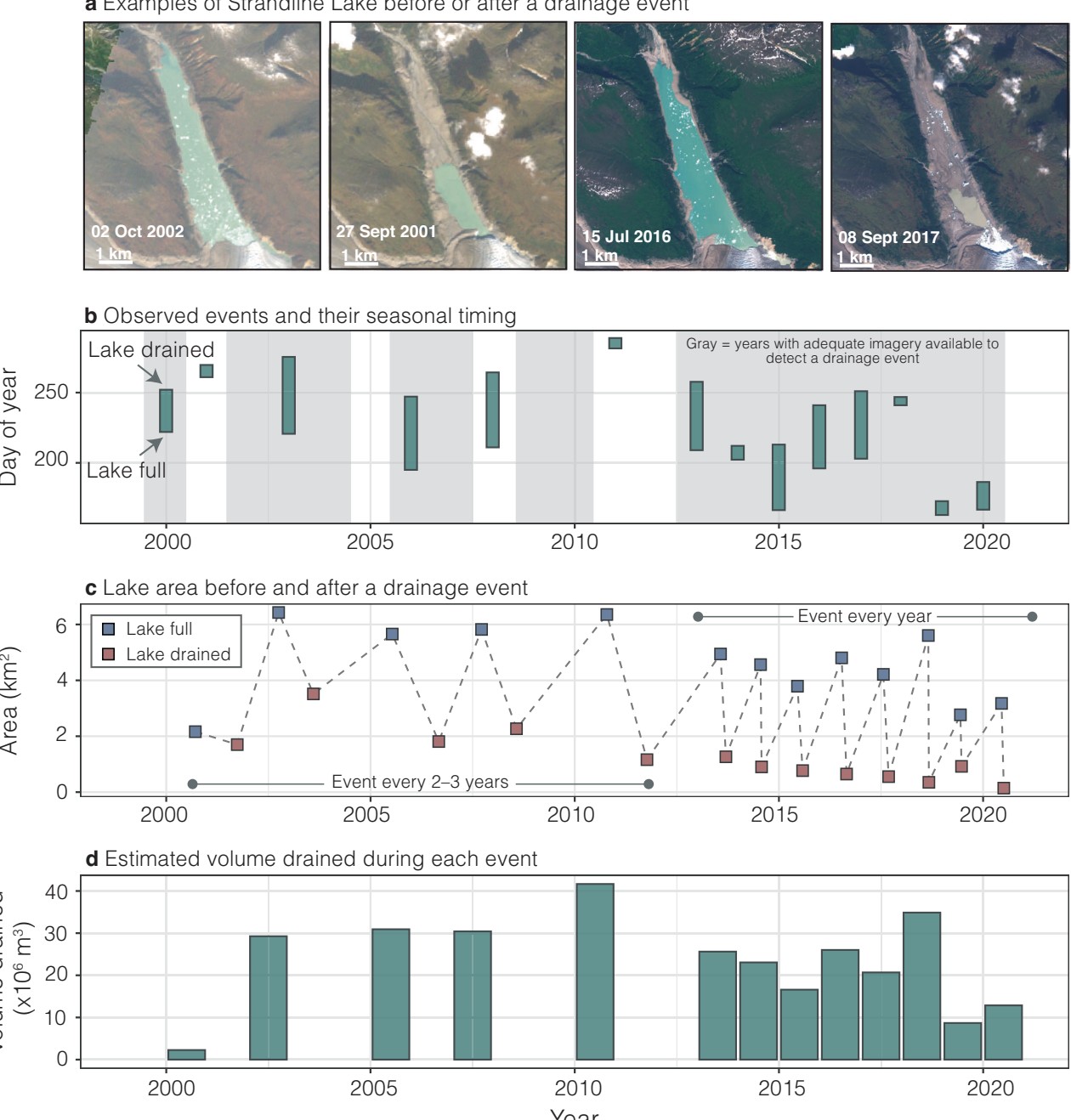

**Fig. 3 | Strandline Lake exhibits trends of increasing frequency of drainage events and decreasing area and volume over time. a** Examples of Sentinel-2 images (Copernicus Sentinel data) when Strandline Lake was full or drained from the early 2000s and mid-2010s. See location of Strandline Lake in Fig. 1. **b** Observed events with the bottom of the segment representing the last image when the lake was full and the top representing the first image the lake was drained, indicating that the lake must have drained between those two dates. Gray bars indicate years with adequate imagery to detect an event. Events which occur in years without a gray bar are detected with imagery from two separate years. **c** Area of the lake before (blue) and after (red) each drainage event. **d** Estimated volume drained for each drainage event.

region[27] will shorten the life cycle[38] of persistent and newly formed ice-dammed lakes.

## The role of dam type in drainage frequency

The documented annual average of ~33 events per year across Alaska between 1985 and 2020 is 12 times greater than the regional rate of GLOFs from ice-dammed lakes documented in High Mountain Asia (average of 2.75 events per year over the same time period[13]). This striking difference in frequency between Alaska and the Himalaya can at least partially be attributed to the abundance of ice-dammed lakes in

Alaska with documented drainages ($n = 106$) as compared to the number of ice-dammed lakes in the Himalaya that have one or more documented drainage events ($n = 21$)[13].

Although moraine-dammed lakes make up the majority of ice-marginal lakes in Alaska, only four GLOFs have been documented from moraine-dammed lakes since 1985[39]. The relatively low frequency of moraine-dammed lake drainage events in Alaska is likely related to the structure and location of Little Ice Age moraines (LIA; ~1350–1850 AD in the Northern Hemisphere[40]). In many instances, LIA moraines are located in large, flat valleys, rather than the steeper settings found

in the Himalaya or Tropical Andes. The triggering mechanism for moraine-dammed lake floods are often ice or rock fall into the lake, which can be triggered by earthquakes or initiate from steep adjacent slopes[17]. Few moraine-dammed lakes in Alaska are currently surrounded by these types of steep, triggering slopes, which reduces the likelihood of moraine-dam failure from such a trigger. Additionally, some moraine dams remain stable due to low-gradient outlet channels draining the lake, maintaining a stable lake level[1]. Other moraine-dammed lakes may slowly infill with sediment (as occurred for the seven lakes lost between 1984–2019[14]), which results in water displacement and a low likelihood of an outburst flood[17].

### Regional decrease in lake volume and estimated peak discharge since the 1950s

For a subset of lakes that had historical aerial imagery available in the ~1950s (n = 33; see Methods), we manually delineated lake perimeters to add an additional 30 years to the record of lake area change. Using an established area-to-volume scaling relationship[9], we estimated lake volumes within each of the three time periods (1950s, 1980s, and 2010s). We then developed a region-specific empirical relationship between lake volume (assuming that lake volume is equal to flood volume) and peak discharge to estimate the potential peak discharge of each lake at each time step (Fig. 4; see Methods and Fig. S10). We found that 70% (n = 85) of the lake population experienced detectable area change between the 1980s and 2010s, with 79% (n = 67) of these lakes decreasing in area, and 61% (n = 52) experiencing a detectable decrease in estimated volume (Figs. S15, S16). Over 50% (n = 44) of the lakes with detectable area change experienced a significant decrease in estimated peak discharge over time, while 19% (n = 16) experienced an increase. The median percent area change was more negative between the 1980s and 2010s than the 1950s and 1980s. This finding is consistent with trends in ice volume loss, with an acceleration of mass loss between 2000 and 2019[27,41]. We also found that all ice dams are thinning, though there is no statistically significant relationship between rate of thinning and rate of lake area change (Fig. S11).

A historical inventory of ice-dammed lakes created with aerial imagery from the 1960s identified what they assessed to be 32 of the most hazardous or unique ice-dammed lakes, based on lake size, location, and historical flood records[15]. At present, 20 of these 32 (63%) lakes remain, though four have transitioned to a more stable dam type (i.e., now bedrock dammed or disconnected from its associated glacier). This means that 16 of 32 ice-dammed lakes remain, implying a 50% reduction in the most hazardous lakes since the 1960s. Of the lakes that remain, nearly all have experienced a decrease in area, likely reducing the maximum peak discharge possible for a modern outburst.

The size and characteristics of the river or lake into which a lake drains, as well as the timing of the drainage, additionally factor into the impact of an event. A large river or proglacial lake can mute the downstream signal of a lake draining[32], and whether the drainage occurs before or after river ice breakup or during low or peak flows can change local hazards. For example, Van Cleve Lake is a large (5.5 km² in 2016–2019) ice-dammed lake which drains through Miles Lake and into the Copper River. Due to the large mean annual flow (~1755 m³/s)[42], these drainage events are presently difficult to detect within the Copper River hydrograph. Conversely, Suicide Basin, dammed by the Mendenhall Glacier, has an area of only 0.5 km² yet impacts homes and roads downstream due to the lake's proximity to settled areas and its large drainage discharge relative to typical flows in the Mendenhall River[19,43].

Determining relationships for event magnitude and downstream impact are critical next steps to fully characterize hazards from glacial lakes, albeit challenging ones[4]. However, this study takes an important step towards a more accurate characterization of ice-dammed lakes across Alaska by determining the frequency, volume, and change of lake drainage events. The observed decrease in ice-dammed lake volume and peak discharge, combined with unchanged frequency of events over time, suggests that the regional GLOF hazards from ice-dammed lakes has decreased, though this trend varies on an individual lake basis. This provides valuable information to aid local agencies and decision-makers in making informed decisions pertaining to water resources and hazard mitigation.

A global compilation of ice-dammed lake drainage events reports 1569 events from 186 lakes between 1900 and 2020[31], whereas our analysis documents 1150 events from 106 lakes between 1985 and 2020 from just within the Alaska region. This suggests that many ice-dammed lake outburst floods have gone undetected globally and warrants systematic study of other regions as well.

Alaska is losing ice mass at one of the fastest rates globally and this rate is accelerating[27], causing profound physical changes to the region's glaciers and downstream environments. This study triples the previous estimate of ice-dammed lake drainage events over the satellite record in Alaska, identifying 1150 drainage events from 106 individual ice-dammed lakes. Nearly 75% of lakes with a statistically significant trend experienced a progressively earlier release date over time. Whereas the frequency of lakes that drain each year between 1985 and 2020 has remained unchanged after accounting for an increase in image availability, the area, volume, and estimated peak discharge of ice-dammed lakes have decreased between the 1950s and 2020. This implies a decrease in the regional GLOF hazard from ice-dammed lakes. Understanding when and where lake drainage events occurred in the past is critical for predicting future lake behavior, with implications for flood hazards, which can have large societal and ecological impacts. This study provides a framework for the systematic and unbiased assessment of ice-dammed lake drainage events that can be applied to other regions around the world.

## Methods

Our study includes ice-dammed lakes within the Randolph Glacier Inventory Region 01 (Alaska and NW Canada) identified in a satellite image-derived ice-marginal lake inventory[14], lakes which remain from a 1960s aerial mage-derived ice-dammed lake inventory[15], as well as lakes with documented drainage events[18]. In total, 121 ice-dammed lakes with an area >0.1 km² were investigated, and 106 of these lakes experienced one or more drainage event.

### Imagery dataset

For each lake, we created a true color timelapse video in Google Earth Engine using all available Landsat 5 (Thematic Mapper, TM), Landsat 7 (Thematic Mapper Plus, ETM+), Landsat 8 (Operational Land Imager, OLI), and Sentinel-2 (Multi-Spectral Instrument, MSI; Copernicus Sentinel data) imagery with less than 20% cloud cover from 30 May to 1 November of each year, from 1985 to 2020. Landsat images courtesy of the U.S. Geological Survey. In total, subsets of more than 14,000 satellite images were analyzed. Over the entire study period, there was an average of 3.4 images per lake per year (min = 0, max = 35). The average number of images available tripled for the 2013–2020 period following the launches of Landsat 8 and Sentinel-2 (Fig. 1c), with an average of 10.6 images per lake per year. Our study is limited by the availability of cloud-free satellite imagery, particularly before the launch of Landsat 8 in 2013 (see Fig. 1c).

For each image, the lake was manually classified as full, partially full, or drained. A drainage event was considered to have occurred between images when the lake was full in one image and partially or fully drained in the subsequent image or a change from partially full to drained (see Fig. S7 for examples). We know the drainage occurred between these two images, though not the precise drainage date (Fig. S8). When the two images (full, drained) are from different years, the event is assigned to the year when the lake was observed drained. In this study, we use the term "drainage event" rather than GLOF, as we

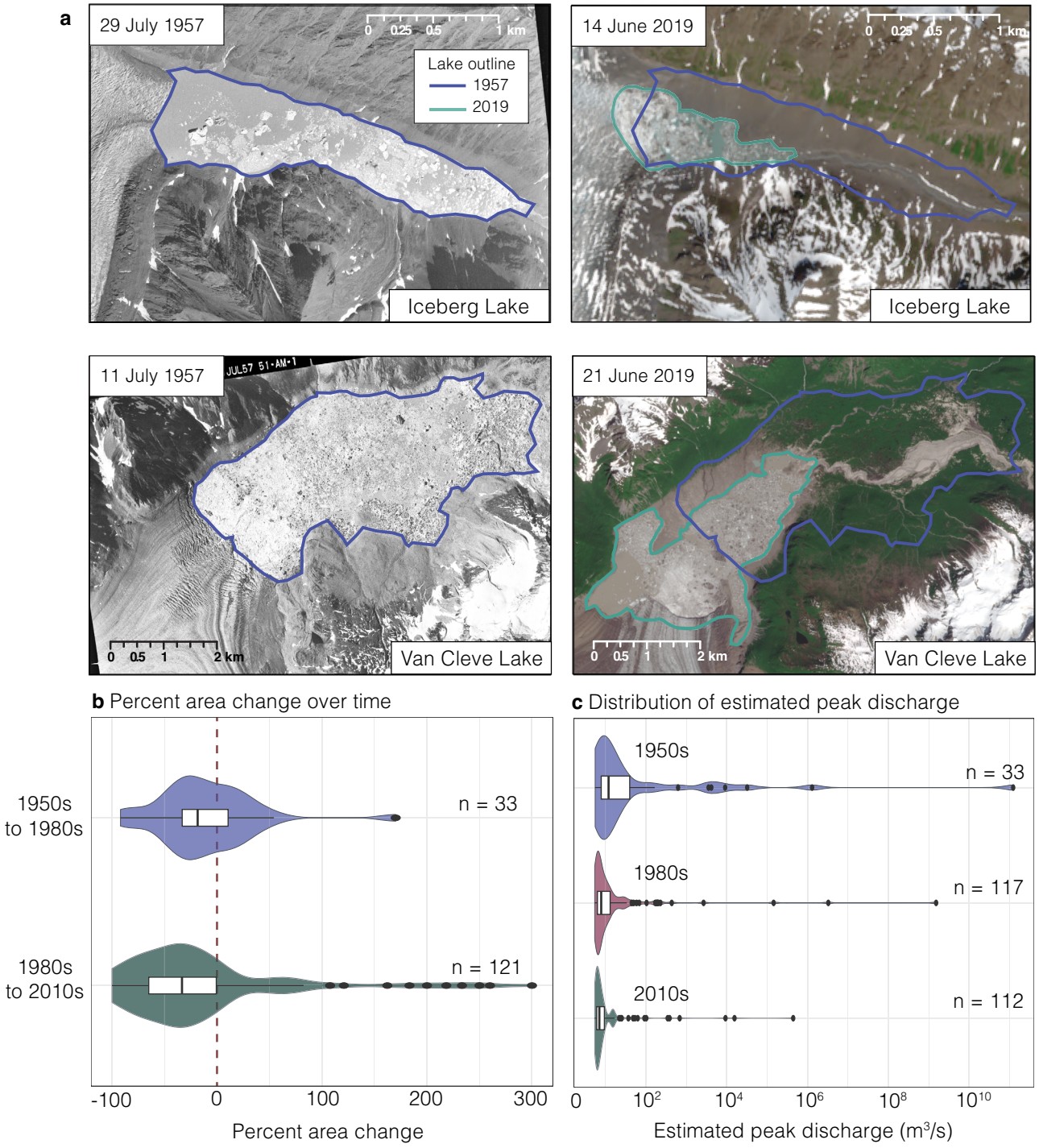

**Fig. 4 | Estimated peak discharge over time. a** Examples of persistent lakes (top is Iceberg Lake- RGI60-01.10778n210 and bottom is Van Cleve lake - RGI60-01.13531n415) with images from the 1950s on the left and modern imagery on the right (courtesy of the U.S. Geological Survey). Purple outlines show lake extent from the 1950s and green outlines show lake extent in 2019. **b** Violin and boxplots show the distribution of percent area change between the 1950s to 1980s (purple) and the 1980s to 2010s (green). **c** Violin and boxplots show the distribution of estimated peak discharge for each time period, displaying a reduction in the largest peak discharges over time. The box encompasses data between the 25th and 75th percentile and the line represents the median discharge.

cannot identify the nature and rate of drainage for any particular event, and the term GLOF implies catastrophic drainage.

### Determining drainage frequency over time

We assessed the change in frequency of events over time using two independent methods. Method one looked at how many lakes had a documented drainage event relative to the number of lakes that had adequate imagery each year, while method two explored

whether a trend existed over time if image availability remained constant.

For method one, we used three different criteria of "adequate imagery" determined using the interquartile range (IQR), ranging from least to most restrictive, to assess whether there was a significant change in drainage frequency over time (Fig. S4). In our record, the distribution of days between images that captured a drainage event indicates an average of 28 days between images, and the empirical

probability distribution indicates 18, 32, and 48 days for the 25%, 50%, and 75% quantiles, respectively (Fig. S3). The least restrictive criteria required that for any given year and any given lake, there had to be at minimum two images that were at least 18 days apart. The medium restrictive criteria required two images that were at least 32 days apart (>50% chance of capturing an event; Fig. S8). The most restrictive criteria included the timing of the images, requiring two images that were at least 32 days apart and their dates bracketing ± 20 days of the median drainage day of year (calculated using the midpoint between date full and date drained for each event per lake). Given the progression in these criteria, the least restrictive criteria resulted in the greatest number of lakes with adequate imagery per year while the most restrictive criteria resulted in the lowest number of lakes with adequate imagery per year. The ratio of number of events detected to number of lakes with adequate imagery per year was used to normalize for image availability in each criteria (Fig. 1d).

For method two, we used a bootstrap method to determine whether we could identify a trend in event frequency if image availability were constant over time. We tested this by randomly selecting the same number of images each year (we tested both 30 images and 50 images per year) between 1985 and 2020, then calculating the number of events captured by those images and fitting a linear trend in the number of events through time. We did this 10,000 times to capture the variability in slope, and reported the 95% confidence interval for the trend in frequency of events over time. We found that if the number of images per year is kept constant, there is no significant trend in frequency over time. This results in an average trend of −0.006 ± 0.001 events per year (Fig. S5), or −0.6 events per century, which is negligible in practical terms. These results support our finding of unchanged frequency of ice-dammed lake drainage events between 1985 and 2020, and that image availability is the biggest driver of the increase in documented events over time.

We also investigated additional characteristics of the ice dams within the Open Global Glacier Model[44]. Dam locations were manually chosen near the center of the ice damming each lake, and values were averaged within a 100 m diameter buffer. We compared statewide mean annual air temperature[45], dam ice thickness[46], and dam thinning rate[27] to see if there are any clear relationships between these variables and the number of events per lake or the change in lake area over time.

### Trend in release timing

Trend in drainage day over time for each individual lake with five or more events ($n = 74$) was assessed by Monte Carlo methods to account for the unknown exact date of release. For each lake, 10,000 simulations were run in which a time series was constructed by randomly selecting a day of year between the last date full and first date drained for each year to be the exact drainage date (using uniform distribution within the window). We then performed linear regressions on the synthetic datasets and recorded the slope of the trend through time. We then assessed the distribution of these 10,000 slopes to determine whether a temporal trend in lake drainage date exists at the 95% confidence level given the variation in the data (Fig. S9). We excluded events in which the two images capturing the event were from different years, as including these types of events and allowing for winter drainages skewed the trends more negative (earlier events over time). Including these events, therefore, likely falsely inflates the number of lakes that had a significant trend in timing over time.

### Lake area, volume, and estimated peak discharge

Lake area and lake area change for the 1980s and 2010s are based on outlines from ref. 14, and more detailed records of lake area for the five largest lakes were manually outlined before and after each drainage event (Figs. S13, S14). Lake perimeters from the 1950s were manually digitized for lakes which had imagery available through Earth Explorer (https://earthexplorer.usgs.gov/). These images were

manually georeferenced and ranged in resolution from 0.3 to 1.5 m pixels. Error in lake delineation was calculated assuming an error of ±0.5 pixel for the entirety of each lake perimeter[47–52],

$$E = P * R \tag{1}$$

where $P$ is the perimeter of the lake (km), $R$ is half a pixel of the imagery (0.015 km for Landsat, 0.0015 km for historical aerial images), and the resulting error ($E$) is in $km^2$.

Lake volume estimates for lakes greater than 0.5 $km^2$ were calculated using the non-linear power scaling described by ref. 9,

$$V = k_1 A^{k_2} \tag{2}$$

where $V$ is lake volume (x $10^6$ $m^3$), $A$ is lake area ($m^2$), coefficient $k_1$ has an estimated value of $4.66 \times 10^{-11}$ and coefficient $k_2$ has an estimated value of 1.76. To account for the uncertainty when calculating the volume of the lake, we used the Monte Carlo method for the lake area as well as the area to volume calculation. We do this by setting a range of values that the area could be (from area − error to area + error), and for each run we take a random number between those two values (using uniform distribution) to use as the area. We also set a range of values for coefficients $k_1$ and $k_2$, determined by the standard error reported by ref. 9. A value was then selected from a normal distribution of the coefficient values for each of the 10,000 runs. We used the IQR of volume estimates to define the most likely range in values for the volume of the lake. Volume ranges were then compared between the time periods (1950s, 1980s, and 2010s; Fig. S15), and if the two ranges did not overlap the lake was determined to have a detectable change in volume.

For lakes less than 0.5 $km^2$, lake volumes were estimated using the log-log-linear scaling described by ref. 9,

$$V = e^{\beta_0} A^{\beta_1} \tag{3}$$

where $V$ is lake volume (x $10^6$ $m^3$), $A$ is lake area ($m^2$), coefficient $\beta_0$ has an estimated value of $4.66 \times 10^{-11}$ and coefficient $\beta_1$ has an estimated value of 1.76. The Monte Carlo method was used again, drawing from a range in lake areas and coefficient values determined by the error.

To create a regional volume to peak discharge relationship, we used an existing database (https://www.weather.gov/aprfc/gdlMain) with a record of select ice-dammed lakes that have documented flood volume and peak discharge ($n = 99$; Fig. S10). As has historically been used[16,53,54], we fit a power-law relationship:

$$Q_{max} = K V^b \tag{4}$$

where $Q_{max}$ is the peak discharge (in $m^3$ per second), $V$ is the volume of water drained (x $10^6$ $m^3$), $K$ has a value of 2.302 (SE = 0.3626), and $b$ has a value of 0.885 (SE = 0.0715). Here, we assume that the volume of water drained is equal to the estimated volume of the lake. For the volume-to-peak discharge calculation, we used the Monte Carlo method for 10,000 runs, incorporating a range of possible volumes (IQR from the volume calculation) and a range in coefficient values determined by the standard error. Estimated peak discharge ranges (IQR), which did not overlap between the time periods, were determined to have a detectable change in peak discharge (Fig. S16).

### Challenges and limitations

This study is limited by satellite imagery availability and the spatial and temporal resolution of this imagery. Due to limited image frequency, we were unable to detect more than one drainage event per year, though lakes such as Lake No Lake and Suicide Basin have experienced multiple drainages in a single year[19,20]. The ice-filled

nature of many ice-dammed lakes also complicates the distinction between a full and drained lake, as sometimes the only indicator that a lake drained is a small depression in the ice floating on the lake surface (Fig. S7). We were also unable to detect if a lake drains after snow cover onset. For example, Snow Lake tends to drain in the fall, often after snow cover onset and has been observed through hydrological methods to drain every 2–3 years since the 1950s[18] (Fig. S17). However, our methods only capture three events since 2015, likely due to the heavily iceberg-filled nature of the lake as well as its drainage timing. The 30 m resolution of Landsat imagery is not sufficient to see small changes in ice topography when the lake drains (Fig. S17), and we are only able to detect Snow Lake events with Sentinel-2 imagery (10 m resolution). Additionally, our use of a 20% cloud cover threshold likely excludes some viable images which may reveal a drainage event. Thus, the reported findings here are a clear minimum of the number of ice-dammed lake drainage events that occurred between 1985 and 2020.

While we estimated peak discharge based on lake volume, we acknowledge that drainage rate impacts event magnitude. For example, a small lake that drains quickly can have a large peak discharge, and conversely, a large lake which drains slowly could have a smaller peak discharge and little to no downstream impacts. We are unable to measure drainage duration as a typical lake drains on the order of hours to days, which is much shorter than the average 28 days between images in our dataset. We also recognize that the relationship between lake volume and peak discharge can be more complicated. For example, the presence of ice within a lake can alter this relationship, decreasing storage capacity yet increasing pressure on the draining water[53]. Considering these limitations, we use estimated peak discharge as a first order estimate of event magnitude to assess changes over time. For lake volume and peak discharge, we use the IQR to identify the most likely range of values for each lake but acknowledge that values may fall outside this range. Therefore, the extreme ends of the distribution may not be captured and could impact hazard assessment.

## Data availability
The full ice-dammed lake drainage event dataset used in this study is available in the Hydroshare database under accession code https://www.hydroshare.org/resource/930f35f1a68949cb9963903b95caadea/.

## Code availability
The Google Earth Engine script used in this study is available at https://www.hydroshare.org/resource/930f35f1a68949cb9963903b95caadea/ in the "AK_Timelapse_GEECode.txt" file.

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

## Acknowledgements
We would like to thank Janet Curran, Crane Johnson, and the Alaska-Pacific River Forecast Center team for insightful discussions. We also thank Jessica Rick, David Rounce, and Jeremy Littell for useful suggestions and assistance. This material is based upon work supported by the National Science Foundation Graduate Research Fellowship awarded to B.R. under Grant No. 006784-00002. D.M. and S.W.M. acknowledge NASA award 80NSSC20K1624. This publication was partially funded by the Alaska Climate Adaptation Science Center.

## Author contributions
All authors contributed to the conceptualization of the study. B.R. and D.M. designed the study. B.R. performed the analysis and prepared the manuscript. D.M., S.W.M., and W.H.A. contributed to data interpretation and provided manuscript revisions.

## Competing interests
The authors declare no competing interests.
