## [Peer Review File · Nature Communications]

Unchanged frequency and decreasing magnitude of outbursts from ice-dammed lakes in AlaskaEditorial Note: Parts of this Peer Review File have been redacted as indicated to remove third-party material where no permission to publish could be obtained.

REVIEWER COMMENTS

Reviewer #1 (Remarks to the Author):

Dear Editors,

Dear Authors,

Thank you for giving me the opportunity to review the manuscript entitled "Decreasing hazards from glacial lakes in Alaska since the 1960s" by Brianna Rick and co-authors (paper NCOMMS-22-30500-T). In this study, Rick et al. compiled a multi-temporal inventory of drainage events (or outbursts) from ice-dammed lakes in Alaska. In their earlier work, the authors examined changes in the number and size of different types of glacial lakes in Alaska. This manuscript is a logical extension of their work and focuses on the drainage of more than 100 ice-dammed lakes. Alaska forms a hotspot in the global distribution of ice-dammed lakes that pose a threat to downstream communities, infrastructure, and habitats. The updated lake drainage inventory exceeds previous surveys for this region by a factor of three. However, the actual number of lake drainage events remains unknown as the detection of ice dam failures depends on the availability of cloud-free Landsat and Sentinel-2 imagery. The authors conclude that the number of lake drainage events detected per available satellite image has remained constant since the mid-1980s. They also examine the timing of lake drainage events, but refrain from a detailed analysis of the magnitude of the outflows, i.e., potential peak flows or flood volumes.

I commend the authors for their efforts in compiling this first multitemporal database of drainage events in ice-dammed lakes in Alaska. Such inventories provide important baseline data for assessing hazards and risks posed by glacial lakes. The article is generally well written, the figures are of good quality, and the text is appropriately referenced. Yet, despite the value of a unique compilation of lakes and potential outbursts in this region, I cannot recommend to publish this study in its present form for four main reasons. I give more detail in the specific comments below.

1) The conclusion of a "decreasing hazard" is only partially supported by the data. The authors assess the frequency of lake drainage events, but hardly the magnitude of these events. As a solution, I recommend that the authors either reword the title and portions of the manuscript to focus consistently on the frequency and timing of lake outflows, or that they conduct a thorough analysis of the magnitude of outflows so that the conclusion can be supported by data.

2) The study could also have a better balance between regional and local scale analysis. For example, trends in drainage frequency are shown at the regional scale (Fig. 1), but are rarely addressed at the local scale (i.e., trends in the number of annual drainage events from individual lakes). In contrast, trends in timing are shown locally but not regionally. Finally, trends in lake area before and after drainage or flood volumes are shown for a handful of lakes, but not for all lakes. It is difficult to say whether the selected cases are adequate to represent the entire lake population.

3) I realize that compiling lake discharge events was already a major effort. That said, I see surprisingly little additional work to explain the observed trends (e.g., earlier timing or smaller lakes). For example, the authors speculate that "we expect lakes that fill and drain to move upward along glaciers"; that decreasing "height of a dam limits its ability to impound water"; or that "lake water temperature, dam strength, and ice dynamics" might explain the frequency of drainage events. Many of these variables are actually freely available to test these hypotheses (e.g., thermal bands in Landsat imagery, gridded data on ice flow velocities, glacier thickness, glacier elevation changes since 2000, lake elevations from DEMs), and I wonder why the authors did not take this step. Such additional analysis would provide a refreshing look at previous statements about biases in GLOF reporting or trends in GLOF timing.

4) From a methodological perspective, I have concerns about whether the 20% threshold in cloud cover might be too rigid (see explanation below) and whether a large number of runoff events have gone still unnoticed in the analysis. This is not a major issue in itself, as the authors make clear that a complete inventory is not possible. However, I would be interested to know if increasing the CC threshold also increases the number of drainage events detected (e.g., for the cases shown in Figure S10). In addition, I would also welcome a validation of the Post and Mayo map, as the authors conclude that the number of lakes has drastically decreased between the creation of this map (in the late 1960s) and the beginning of the Landsat era. Post and Mayo's map has been used repeatedly in previous studies to claim a decline in the number of ice-dammed lakes. However, I find it somewhat questionable that the number of lakes should have decreased by about two-thirds in just two decades, while it remained more or less unchanged in the following years.

In summary, I think this manuscript could deserve a better balance between strengthening the focus of the research goals, the analysis of potential drivers of observed trends, and a thorough validation of their core conclusions (i.e. decreasing hazard = frequency AND size of lakes). I would be delighted to look at a revised version.

Specific comments

L1 – title: Not sure whether ‘decreasing hazards’ appropriately address the core findings of the paper. Hazard is a function of the frequency and magnitude of a given process. I agree that the authors thoroughly studied the frequency, but the magnitudes are only discussed, i.e. there is no estimate of flood volume or peak discharges of the reported events. Suggest to emphasize the observed trends in frequency, associated biases, or the changes in drainage timing in the title.

L14: ‘climate change’: please be more specific – what is changing that causes ice to retreat? Suggest using ‘atmospheric warming’.

L15: ‘many’ researchers?

L21: ‘apparent’: please be more specific. How did the frequency of detected drainage events change in the study period?

L22: ‘loss’ of? Number, area, volume?

L23-24: This conclusion is only valid if floods from the remaining lakes did not become larger at the same time (which is not analysed systematically in this study). Suggest to tone down this conclusion.

L32: What is meant with ‘sustainability of communities’? Why do GLOFs only impact ‘large’ infrastructure projects?

L33: It would be good to say clearly, either globally or using regional examples, how much the area and number of glacial lakes has changed in a given period.

L34 and throughout the manuscript: please make sure that ‘frequency’ and ‘hazard’ are not used interchangeably. I’m sure the authors know the difference, but readers may get the impression that an increasing frequency needs to increase hazard, which is not always true (especially with the data shown in this study).

L44: ‘repair’ – maybe more specifically ‘close en- or subglacial tunnels’?

L47: or Suicide Lake. It’s ‘Lake No Lake’, see reference Geertsema and Clague (2005). ‘moraine-dammed’ (add hyphen)

L48: ‘damage’: perhaps use ‘incision’, ‘breach’ or ‘failure’?

L53: ‘contemporary’ lake population? Or which period do these percentages refer to?

L58: ‘pose the greatest hazard’: conjectural without further information, tone down.

L64: The reason why the timing of drainage is important could be motivated more, not least because much of the results focus on it.

L72: ‘70%’ is this the same number as the 75% in the abstract?

L71-77: Paragraph seems to slightly be out place. I suggest swapping the order of this and the

previous paragraph.

L79-83: Reference to methods is missing. Even though this is a short paper, readers demand more information in the main text to assess how the authors have ensured that ice-dammed lake drainage events are reliably identified. In other words, the remote-sensing analysis needs to be at least briefly introduced here. 'size' is only assessed for case studies a not in a comparable detail as 'frequency and timing'?

L81-83: 'as they dominate ... in Alaska': repetition, consider deleting.

L87-90: 'Ice-dammed lakes ... of hazards': again repetition. On the other hand, this motivation could give the impression that the authors have calculated return intervals or flood volumes, which is not the core of this paper.

L94, L97: Unclear how the authors define 'suitable' and 'adequate'.

L105-109: Errors are missing in the trends. This entire approach of bias correction (i.e. the different restrictive criteria) is very hard to follow in the present form.

L113-115: Interesting finding, could the authors also express this in absolute numbers? It is striking that the number of lakes decreased by two thirds between 1970 and 1985, but remained almost constant between 1985 and 2020. Could it be that Post and Mayo's map includes hypothetical lakes, i.e., basins that might be suitable for trapping meltwater but were not checked to see if a lake had actually formed there? I have a hard time understanding the underlying trends in the number of lakes, given that glacier retreat rates have been exceptionally high in the 21st century, but the number of lakes has remained fairly constant. Please discuss this issue in more detail.

Fig. 1: a: needs a grid with latitude and longitude. Some more labels of cities or mountain ranges could be helpful. Lake area refers to the size in 2020? e: how do the authors explain the missing events in roughly 1990? If the authors multiply a constant rate with an (almost constant) number of lakes, then the actual number of drainage events needs to remain constant as well, no?

L135: where do Figs. 2 and 3 show frequency?

L136: 'over half' means 61%?

L139: The global trends in GLOF timing seem to contradict the findings in this study, no? The authors found that 61% of all lakes had no change in timing, while globally two thirds showed a change.

L141: less 'hydrostatic' pressure, because lake depth decreases. 'destabilize': maybe also add 'to lift' or 'float' the ice dam?

L143: Add reference on changing snow melt regimes?

Fig. 2: RGI subregions are not shown. Add to Fig. 1? Please avoid using red and green in the same figure. How do the authors calculate trends for lakes that had two drainage events? This trend should have zero variance.

L153: Which figure shows this 'notable' pattern? How many lakes had an increasing annual drainage frequency? Hard to tell whether the two examples (Strandline and Summit Lake) are suitable to reflect the bigger picture. One could also argue that the use of such case studies also introduces bias by obscuring much of the underlying variance in the data. Instead, I recommend that the authors perform the same analysis for discharge frequency and volume per lake as they did for timing.

L188-200: Why is there so much focus on moraine-dammed lakes in this paragraph? High Mountain Asia and the Andes also have ice-dammed lakes. Is drainage frequency and timing of ice-dammed lakes different in these regions? I suggest to focus more on drivers (and comparisons) of ice-dam failures, rather than shifting the focus to moraine-dam failures, which were not the main subject of this study.

L188-189: How did the authors ensure that the rate of moraine-dam failures is free from reporting biases?

L192-193: Is there any study that has systematically explored the nexus between valley shape and the abundance of moraines?

L211: Does this mean that smaller lakes are more short-lived? Or that they are more difficult to detect? Please discuss possible causes for this observation.

Fig 4b: Is there any evidence of old strandlines in the empty lake basins? This could help to support the hypothesis of the rapid disappearance of lakes since the late 1960s (which I am still somewhat skeptical of).

L228-229: Is this a subjective or objective definition of hazard? In other words, how did the return levels for these lakes change?

L234: Be careful with this statement. The study by Jenson et al. (2022, The Cryosphere) showed that lakes can maintain high peak discharges even when lake volumes decline.

L237: Please add a reference that systematically assessed this scaling relationship.

L236-247: Why do the authors refrain from showing and discussing reported trends in peak discharge? Arguing with trends derived for a number of lakes would enrich this rather qualitative paragraph. I guess they are aware of this compilation? <https://www.weather.gov/aprfc/gdlMain>

L259-260: 'critical next steps to fully characterize hazards from glacial lakes': the title of this manuscript suggests that you already solved this problem? Again, I suggest to tone down the focus on hazard in this paper.

L270-281 – Summary: largely repetition of the Abstract, and adds not much to the previous paragraphs.

L273: what is 'dam integrity'?

L278: 'dramatically': please remain subjective.

L377: Why did the authors choose such a low threshold for cloud cover? In many satellite images along the coast of Alaska, I have seen that the ocean is covered by clouds while the mountains are cloud-free. Even with a threshold of 80-90%, parts of the image, and thus the lakes of interest, may still be cloud-free and allow measuring the frequency and approximating the timing of lake drainage. A higher threshold could be especially helpful in the early years of the study period, when only Landsat TM is available. In this context, the question arises why the authors did not use Landsat MSS imagery. These go back to 1972 and would make it possible to verify the lakes indicated in the Post and Mayo map.

L385-392: Why did the authors refrain from mapping lakes areas? This information would add so much to the discussion of changing hazards. I also understand the term 'drainage event', but the authors could also evaluate downstream changes (erosion, aggradation) and river avulsions to determine if the drainage event was indeed rapid or catastrophic? Slow drainage would otherwise undermine the motivation that the hazard from ice-dammed lakes in Alaska is decreasing.

L416-421: Which likelihood function did the authors choose? The outcome 'day' needs to take values between 0 and 365 and demands a response other than the Gaussian.

Reviewer #2 (Remarks to the Author):

The reviewed manuscript presents a retrospective analysis on the decreasing hazards from ice-dammed glacial lakes in Alaska over the past decades. This work was largely based on various remote sensing datasets and the GEE platform. The authors note a lack of systematic and spatially explicit observations of lake drainage events in Alaska and suggest an unbiased look at other cold regions, which does highlight the potential importance of such work. The results of this study would be of interest to the global GLOF research community and hazard policymakers. However, the manuscript lacks in-depth analysis and discussion on the drivers and impacts of ice-dammed GLOFs and the produced inventory could be further enhanced by adding more useful parameters. For the reasons above (as detailed below) I would recommend a moderate to major revision.

Major comments:

Introduction: GLOF drivers should be briefly mentioned in the introduction including mass movements into glacial lakes, extreme rainfall events, glacier surges (especially the case for ice-dammed lakes), seismic activity, and piping failures. Also, GLOF impacts on infrastructure and livelihoods could be more explicitly discussed (e.g., for downstream hydropower systems, extensive fluvial erosion and secondary landslides caused; see Li et al., 2022)

Line 79 and the GLOF inventory: I checked the dataset and found that the inventory only includes dates (full/empty) and longitudes and latitudes of glacial lakes. It would be more helpful to include the information for the lake area and also drivers and impacts of these lake drainage events. This could be challenging to detail all the 1150 events, but such information would indeed be more useful for the science community and policymakers.

Line 96-99: I feel that the simple correlation between detected GLOF events and imagery number does not sufficiently and explicitly demonstrate that increased image number is the sole driver of the increased reported GLOFs. The increased satellite images could be the key driver, but other factors like glacier surges and climate drivers should also be analyzed. An in-depth analysis of the GLOF drivers from a process perspective would enhance the argument.

Discussion: I find that glacier surge as a potentially important driver of ice-dammed lake outburst floods was not sufficiently discussed in the paper.

Minor comments:

Title: maybe it should explicitly mention the ice-dammed glacial lakes - the focus of this study. Also, it should be 1980s or 1960s?

Line 32: it is unclear what do you mean by the 'large infrastructure projects. hydropower projects or highways?

Line 46-47: similarly, outburst floods from the ice-dammed Lake Kyagar and Lake Merzbacher Lake in High Mountain Asia also occur more frequently, as compared to the moraine-dammed lakes (see Li et al., 2022)

Line 56: 330 drainage events?

Figs. 1-3: is it possible to also report the uncertainties associated with these frequencies and timing, areas, and volumes?

Fig. 3: please add the date of the images and scale information in fig. 3a; the location of this Strandline Lake should be marked in Fig.1 to facilitate the readers

Line 181-184: it seems not fair to compare with the moraine-dammed lakes in the Himalaya. Instead, it should compare with the ice-dammed lakes in the Karakoram (see Bazai et al., 2021)

Line 198-200: This could be true as the sediment supplies from retreating glaciers worldwide have been increasing since the 1950s (see Li et al., 2021)

Line 228: please indicate the time period of the lake inventory reported by Post and Mayo. Did it cover the 1960s?

Lines 259-266: please consider adding a few sentences about the broad implications of this study (e.g. for future infrastructure development and cryospheric hazard mitigation; see also Emmer et al., 2022)

References mentioned:

Bazai, N. A., Cui, P., Carling, P. A., Wang, H., Hassan, J., Liu, D.,... Jin, W. (2021). Increasing glacial lake outburst flood hazard in response to surge glaciers in the Karakoram. *Earth-Science Reviews*, 212, 103432.

Emmer, A., Allen, S. K., Carey, M., Frey, H., Huggel, C., Korup, O., ... & Yde, J. C. (2022). Progress and challenges in glacial lake outburst flood research (2017–2021): a research community perspective. *Natural Hazards and Earth System Sciences*, 22, 3041–3061.

Li, D., Lu, X., Overeem, I., Walling, D. E., Syvitski, J., Kettner, A. J.,... Zhang, T. (2021). Exceptional increases in fluvial sediment fluxes in a warmer and wetter High Mountain Asia. *Science*, 374(6567), 599-603.

Li, D., Lu, X., Walling, D. E., Zhang, T., Steiner, J. F., Wasson, R. J.,... Bolch, T. (2022). High Mountain Asia hydropower systems threatened by climate-driven landscape instability. *Nature Geoscience*, 15, 520-530.

Dear Referees,

Thank you for your helpful and constructive comments. We sincerely appreciate the time, effort, and thought that you put into helping us improve our manuscript.

We have addressed your comments and concerns in detail, with three major changes:

(1) We added additional analysis estimating both lake volume and peak discharge for three different time periods (1950s, 1980s, and 2010s) using an established area-volume scaling and a regionally-calibrated volume-peak discharge relationship to assess the changes in magnitude of ice-dammed lake drainage events over time. We believe this makes the hazards angle of our study even stronger, showing that while the frequency of outburst floods has not changed between 1985 and 2020, the magnitude of these events has substantially decreased over time.

(2) Reviewer 1 prompted us to better validate the Post and Mayo (1971) inventory of ice-dammed lakes from the 1960s. Upon closer inspection, we found the dataset is not appropriate for direct comparison with modern numbers of ice-dammed lakes due to methodological differences (e.g., the historical dataset utilized geomorphic evidence for lake existence while our modern dataset only includes a lake with direct observational evidence). To address this discrepancy, we focused only on lakes which existed in both the 1960s and 1980s, and analyzed imagery from the 1950s to produce lake areas at this time step with consistent methodology. Our use of the 1950s imagery was driven by image availability in public databases.

(3) As suggested, we investigated drivers of the observed changes, and found no clear relationship between ice-dammed lake changes and glacier thinning rate, presence/absence of surging glaciers, or lake area. We believe this demonstrates the complexity of these ice-dammed lake systems and warrants further study to characterize the drivers of changes. We have added text and figures demonstrating these results in the section “Regional decrease in lake volume and estimated peak discharge since the 1950s” and Figure S10.

A recent study published in *Nature* (Veh et al., 2023, [doi:10.1038/s41586-022-05642-9](https://doi.org/10.1038/s41586-022-05642-9)) examines ice-dammed lakes on a global scale using a compilation of documented events. They find a decrease in the peak discharge of the largest 10% of ice-dammed lakes and a shift towards earlier outbursts, which corroborates our findings. Importantly, however, our findings for the Alaska region are based on an unprecedented 1150 events from 121 lakes, while the Veh et al. (2023) global compilation of documented events only contains 1569 failures from 186 lakes, suggesting that many thousands more have gone undetected globally. This highlights the need for more systematic and focused studies, and we provide the framework to systematically and unbiasedly study ice-dammed lake drainage events in other regions around the world. We additionally provide an important assessment of the influence of increasing satellite imagery availability, and caution that future studies must take this into account when assessing changes in frequency over time.

Thank you again for your time and feedback.

Sincerely,

Brianna Rick
Alaska Climate Adaptation Science Center

On behalf of co-authors: Daniel McGrath, Scott McCoy, and William Armstrong

Dear Referee 1,

Thank you for taking the time to review our study and provide thorough and constructive feedback. Your suggestions have improved the quality and clarity of our manuscript. Below we provide detailed responses to each of your comments, with our responses in blue.

Reviewer #1 (Remarks to the Author):

Dear Editors,
Dear Authors,

Thank you for giving me the opportunity to review the manuscript entitled "Decreasing hazards from glacial lakes in Alaska since the 1960s" by Brianna Rick and co-authors (paper NCOMMS-22-30500-T). In this study, Rick et al. compiled a multi-temporal inventory of drainage events (or outbursts) from ice-dammed lakes in Alaska. In their earlier work, the authors examined changes in the number and size of different types of glacial lakes in Alaska. This manuscript is a logical extension of their work and focuses on the drainage of more than 100 ice-dammed lakes. Alaska forms a hotspot in the global distribution of ice-dammed lakes that pose a threat to downstream communities, infrastructure, and habitats. The updated lake drainage inventory exceeds previous surveys for this region by a factor of three. However, the actual number of lake drainage events remains unknown as the detection of ice dam failures depends on the availability of cloud-free Landsat and Sentinel-2 imagery. The authors conclude that the number of lake drainage events detected per available satellite image has remained constant since the mid-1980s. They also examine the timing of lake drainage events, but refrain from a detailed analysis of the magnitude of the outflows, i.e., potential peak flows or flood volumes.

I commend the authors for their efforts in compiling this first multitemporal database of drainage events in ice-dammed lakes in Alaska. Such inventories provide important baseline data for assessing hazards and risks posed by glacial lakes. The article is generally well written, the figures are of good quality, and the text is appropriately referenced. Yet, despite the value of a unique compilation of lakes and potential outbursts in this region, I cannot recommend to publish this study in its present form for four main reasons. I give more detail in the specific comments below.

1) The conclusion of a "decreasing hazard" is only partially supported by the data. The authors assess the frequency of lake drainage events, but hardly the magnitude of these events. As a solution, I recommend that the authors either reword the title and portions of the manuscript to focus consistently on the frequency and timing of lake outflows, or that they conduct a thorough analysis of the magnitude of outflows so that the conclusion can be supported by data.

Thank you for this suggestion. We have changed text throughout to appropriately convey that we are not directly measuring hazards, but rather found an unchanged frequency of events yet a decrease in the magnitude of events, from which hazards can be inferred. We added an analysis estimating peak discharge based on lake volume at three different time periods (where possible). We use an area to volume scaling relationship (from Shugar et al., 2021), and then calculate estimated peak discharge using a regional relationship based on events (n=99) with documented flood volume and peak discharge (<https://www.weather.gov/aprfc/gdlMain>). We believe this new analysis strengthens this paper and allows us to infer a regional decrease in hazards for the studied time period.

2) The study could also have a better balance between regional and local scale analysis. For example, trends in drainage frequency are shown at the regional scale (Fig. 1), but are rarely addressed at the local scale (i.e., trends in the number of annual drainage events from individual lakes). In contrast, trends in timing are shown locally but not regionally. Finally, trends in lake area before and after drainage or flood volumes are shown for a handful of lakes, but not for all lakes. It is difficult to say whether the selected cases are adequate to represent the entire lake population.

Thank you for this observation. To increase the consistency of our results, we updated the main text to primarily focus on subregional results, and moved individual lake results to the supplement. In response to your suggestion, our new analysis of estimated lake volumes and peak discharges helps to address changes in magnitude for each

individual lake. We provide examples to show the complexity and variation in individual lakes, rather than as a representation of all lakes in the region.

3) I realize that compiling lake discharge events was already a major effort. That said, I see surprisingly little additional work to explain the observed trends (e.g., earlier timing or smaller lakes). For example, the authors speculate that "we expect lakes that fill and drain to move upward along glaciers"; that decreasing "height of a dam limits its ability to impound water"; or that "lake water temperature, dam strength, and ice dynamics" might explain the frequency of drainage events. Many of these variables are actually freely available to test these hypotheses (e.g., thermal bands in Landsat imagery, gridded data on ice flow velocities, glacier thickness, glacier elevation changes since 2000, lake elevations from DEMs), and I wonder why the authors did not take this step. Such additional analysis would provide a refreshing look at previous statements about biases in GLOF reporting or trends in GLOF timing.

Thank you for this suggestion. We did investigate these suggested datasets (elevation, ice thinning rates, glacier thickness) and did not find any clear trends, apart from that all lakes are associated with thinning glaciers, though there is not a simple relationship between thinning rate and rate of lake area change. While we agree that an important next step is to understand why we are seeing these changes (i.e. earlier timing, smaller lakes and peak discharge), this analysis is outside the scope of this study.

4) From a methodological perspective, I have concerns about whether the 20% threshold in cloud cover might be too rigid (see explanation below) and whether a large number of runoff events have gone still unnoticed in the analysis. This is not a major issue in itself, as the authors make clear that a complete inventory is not possible. However, I would be interested to know if increasing the CC threshold also increases the number of drainage events detected (e.g., for the cases shown in Figure S10). In addition, I would also welcome a validation of the Post and Mayo map, as the authors conclude that the number of lakes has drastically decreased between the creation of this map (in the late 1960s) and the beginning of the Landsat era. Post and Mayo's map has been used repeatedly in previous studies to claim a decline in the number of ice-dammed lakes. However, I find it somewhat questionable that the number of lakes should have decreased by about two-thirds in just two decades, while it remained more or less unchanged in the following years.

We appreciate the constructive critique about our cloud cover threshold. Before deciding on 20%, we tried a variety of thresholds and chose 20% as a way to maximize the number of good images and minimize the number of unusable images in our dataset. Even at 20% cloud cover, 42% of the images had to be manually removed due to poor quality, and loosening the cloud cover threshold would significantly increase the number of poor images within the dataset.

In order to address the concern with the Post and Mayo dataset (which we agree should be validated in order to have confidence in changes over time), we decided to only look at persistent lakes (i.e. lakes that are present in Post and Mayo and our 1980s/2010s dataset) and their change in area over time, rather than the regional number of ice-dammed lakes. Additionally, we digitized our own lake area outlines using imagery mainly from the 1950s as there is substantial uncertainty in utilizing the Post and Mayo map, which led to a consistent overestimation of lake areas.

In summary, I think this manuscript could deserve a better balance between strengthening the focus of the research goals, the analysis of potential drivers of observed trends, and a thorough validation of their core conclusions (i.e. decreasing hazard = frequency AND size of lakes). I would be delighted to look at a revised version.

Thank you again for these thoughtful and constructive comments.

Specific comments

L1 – title: Not sure whether 'decreasing hazards' appropriately address the core findings of the paper. Hazard is a function of the frequency and magnitude of a given process. I agree that the authors thoroughly studied the frequency, but the magnitudes are only discussed, i.e. there is no estimate of flood volume or peak discharges of the

reported events. Suggest to emphasize the observed trends in frequency, associated biases, or the changes in drainage timing in the title.

Thank you for making this observation. We believe that the addition of our peak discharge analysis allows us to infer that there have been decreasing hazards (unchanged frequency + decreasing size). We have generally toned down the language of our paper and believe it now more accurately reflects the main findings of this study.

L14: ‘climate change’: please be more specific – what is changing that causes ice to retreat? Suggest using ‘atmospheric warming’.

Thank you for this suggestion. We have changed the text to read:
“As glaciers retreat due to atmospheric warming, glacial lakes can form and grow.”

L15: ‘many’ researchers?

Changed to “some” researchers.

L21: ‘apparent’: please be more specific. How did the frequency of detected drainage events change in the study period?

We have modified the text for clarification:

“Using multitemporal satellite imagery, we documented 1150 drainages from 106 lakes over 1985–2020, with an increase in detected event frequency over this interval.”

L22: ‘loss’ of? Number, area, volume?

We replaced the text with the new results showing that 70% of lakes decreased in volume and potential peak discharge since the 1960s:

“Furthermore, 70% of ice-dammed lakes decreased in estimated volume and peak discharge since the 1950s, and nearly a third of lakes released earlier through time.”

L23-24: This conclusion is only valid if floods from the remaining lakes did not become larger at the same time (which is not analysed systematically in this study). Suggest to tone down this conclusion.

With the new results showing a decrease in peak discharge, we believe this conclusion is still valid.

L32: What is meant with ‘sustainability of communities’? Why do GLOFs only impact ‘large’ infrastructure projects?

We have changed the wording for clarification:

“The number of GLOFs per year that a region experiences is a fundamental measure of GLOF hazard, with a change in frequency and magnitude potentially impacting downstream communities. GLOFs can cause devastating impacts to infrastructure (i.e. roads, bridges, hydropower systems), ecosystems (i.e. altering sediment flux, salmon habitat), and livelihoods (i.e. fishing, agriculture)³⁻⁷.”

L33: It would be good to say clearly, either globally or using regional examples, how much the area and number of glacial lakes has changed in a given period.

Thank you for this suggestion. We added the increase in number and total area of glacial lakes globally, and it now reads:

“Glacial lakes have been rapidly increasing in number (53%) and total area (51) globally between 1990 and 2018:”

L34 and throughout the manuscript: please make sure that ‘frequency’ and ‘hazard’ are not used interchangeably. I’m sure the authors know the difference, but readers may get the impression that an increasing frequency needs to increase hazard, which is not always true (especially with the data shown in this study).

Thank you for pointing out this clarification - we have tried to clarify to make this difference clearer to the reader throughout the text. We tried to generally tone down our discussion of hazards and stick to frequency and size.

L44: ‘repair’ – maybe more specifically ‘close en- or subglacial tunnels’?

Thank you for this suggestion - we have added this text for clarification.

L47: or Suicide Lake. It’s ‘Lake No Lake’, see reference Geertsema and Clague (2005). ‘moraine-dammed’ (add hyphen).

The proper lake name, Suicide Basin example, and hyphen have all been added to the text.

L48: ‘damage’: perhaps use ‘incision’, ‘breach’ or ‘failure’?

We changed the text to now read:

“In contrast, moraine-dammed lakes typically drain only once as permanent dam failure is common in the drainage process.”

L53: ‘contemporary’ lake population? Or which period do these percentages refer to?

Yes, this refers to the modern lake population – we have added text for clarification.

L58: ‘pose the greatest hazard’: conjectural without further information, tone down.

Thank you for this suggestion, we have toned down the language.

L64: The reason why the timing of drainage is important could be motivated more, not least because much of the results focus on it.

Thank you for this suggestion. We have added a paragraph describing the importance of timing:

“The timing of when a lake releases water can influence downstream impacts, depending on the conditions of the stream or river into which it drains. If river ice is still present the additional water may cause enhanced flooding, whereas if the river stage (or discharge) is low the lake drainage may have little flooding impact. The timing can also exacerbate ecological impacts; for example, streambed scour can disturb salmon eggs within streambed gravels and decrease wild salmon productivity^{21,22}.”

L72: ‘70%’ is this the same number as the 75% in the abstract?

We have changed the text to reflect the results of Wolfe et al. (2008), which investigated a subregion of Alaska.

L71-77: Paragraph seems to slightly be out place. I suggest swapping the order of this and the previous paragraph.

Swapping these paragraphs does create a better flow- thank you for the suggestion.

L79-83: Reference to methods is missing. Even though this is a short paper, readers demand more information in the main text to assess how the authors have ensured that ice-dammed lake drainage events are reliably identified. In other words, the remote-sensing analysis needs to be at least briefly introduced here. ‘size’ is only assessed for case studies a not in a comparable detail as ‘frequency and timing’?

We added a reference to the methods in the main text:

L92-100: “Focusing solely on ice-dammed lakes, we documented all identifiable lake drainage events over 1985 – 2020 in the Alaska region (including a small portion of NW Canada) using timelapse videos of satellite imagery (Landsat and Sentinel-2) produced in Google Earth Engine (see Methods). We manually analyzed images to determine robust measures of drainage event frequency, timing, and size. We estimated lake volume and potential peak discharge for three different time periods (1950s, 1980s, and 2010s) using empirical scaling relationships. This study is the first to systematically and comprehensively document ice-dammed lake drainage events in the Alaska region and provides an assessment for how these events have changed over time, as well as a baseline and framework for future regional studies.”

L81-83: ‘as they dominate ... in Alaska’: repetition, consider deleting.

This sentence has been deleted. Thank you for the suggestion.

L87-90: ‘Ice-dammed lakes ... of hazards’: again repetition. On the other hand, this motivation could give the impression that the authors have calculated return intervals or flood volumes, which is not the core of this paper.

We have now calculated flood volumes/estimated peak discharge and feel that this sentence is appropriate. We reduced mention of hazards in previous paragraphs to reduce repetition.

L94, L97: Unclear how the authors define ‘suitable’ and ‘adequate’.

We have changed text for clarification – we removed “suitable” and “adequate”, focusing just on imagery availability.

L105-109: Errors are missing in the trends. This entire approach of bias correction (i.e. the different restrictive criteria) is very hard to follow in the present form.

We added errors to the trends and tried to simplify the text for clarity. We also added an independent bootstrapping method that confirms our conclusion in no trend in frequency over time after accounting for imagery availability.

L113-115: Interesting finding, could the authors also express this in absolute numbers? It is striking that the number of lakes decreased by two thirds between 1970 and 1985, but remained almost constant between 1985 and 2020. Could it be that Post and Mayo's map includes hypothetical lakes, i.e., basins that might be suitable for trapping meltwater but were not checked to see if a lake had actually formed there? I have a hard time understanding the underlying trends in the number of lakes, given that glacier retreat rates have been exceptionally high in the 21st century, but the number of lakes has remained fairly constant. Please discuss this issue in more detail.

Thank you for this observation. After a closer look, we agree that we cannot use the Post and Mayo dataset without validation to assess the change in number of lakes over time. Additionally, they used a different criteria for defining a lake – any basin that had evidence of a lake in the previous decade was considered a lake, which could be why they had such a high count of ice-dammed lakes. Instead, we look at lakes that were present in both the 1960s and the 1985-2020 time period, and use aerial images from the 1950s (what is available on Earth Explorer) to manually digitize lake outlines to assess lake area, rather than lake number, over time. With these new areas, we find that the rate of lake area change is greater between 1985 and 2020 than between the 1950s and 1985, which follows the trend in glacier thinning/retreat rates.

Fig. 1: a: needs a grid with latitude and longitude. Some more labels of cities or mountain ranges could be helpful. Lake area refers to the size in 2020? e: how do the authors explain the missing events in roughly 1990? If the authors multiply a constant rate with an (almost constant) number of lakes, then the actual number of drainage events needs to remain constant as well, no?

We added a grid, RGI subregions, and cities to help with orientation. We also clarify that we are showing “contemporary lake area”. We have removed Fig. 1e.

L135: where do Figs. 2 and 3 show frequency?

We have changed this to reference Figs 1, 2, S7, and S8.

L136: ‘over half’ means 61%?

Technically 61% is over half, but we have added the specific number to the text to clear up any confusion.

L139: The global trends in GLOF timing seem to contradict the findings in this study, no? The authors found that 61% of all lakes had no change in timing, while globally two thirds showed a change.

For the 61% percent, all we can say is that we can not identify a trend, not necessarily that there is no trend. This may be because there are too few events to detect a trend, or the range between date full and date empty is too large to identify a trend, but if we had the exact release date we would see a trend over time. Additionally, the global trend only looks at lakes with 5 or more events, whereas we allow lakes with more than 2 events to have a calculated trend.

L141: less ‘hydrostatic’ pressure, because lake depth decreases. ‘destabilize’: maybe also add ‘to lift’ or ‘float’ the ice dam?

We have added these text suggestions for clarification.

L143: Add reference on changing snow melt regimes?

We have added a reference to changing snow melt regimes.

Fig. 2: RGI subregions are not shown. Add to Fig. 1? Please avoid using red and green in the same figure. How do the authors calculate trends for lakes that had two drainage events? This trend should have zero variance.

We added RGI subregions to Fig. 1 and added a map to Fig. 2 to help as well. We changed the color scheme to red, gray, and blue to avoid red and green in the same figure. The variance comes from our method of calculating the trend – since we only know the date the lake was full and the date the lake was empty, we sample any date between those two to possibly be the date of drainage. This allows for variation in the data as we sample 1000 times per lake.

L153: Which figure shows this ‘notable’ pattern? How many lakes had an increasing annual drainage frequency? Hard to tell whether the two examples (Strandline and Summit Lake) are suitable to reflect the bigger picture. One could also argue that the use of such case studies also introduces bias by obscuring much of the underlying variance in the data. Instead, I recommend that the authors perform the same analysis for discharge frequency and volume per lake as they did for timing.

We have now included a reference to Fig. S6 which shows all events for all individual lakes.

L188-200: Why is there so much focus on moraine-dammed lakes in this paragraph? High Mountain Asia and the Andes also have ice-dammed lakes. Is drainage frequency and timing of ice-dammed lakes different in these

regions? I suggest to focus more on drivers (and comparisons) of ice-dam failures, rather than shifting the focus to moraine-dam failures, which were not the main subject of this study.

We changed the first paragraph in this section to focus on ice-dammed lakes, and comparing between regions and their number of lakes that drain as well as drainage rate. We believe the discussion of moraine-dammed lakes is valuable, as many other regions focus on moraine-dammed lakes, and it is unique that in Alaska these types of lakes are not as important for hazards as, say, in the Himalaya. Readers may wonder why we focus on ice-dammed lakes, or why moraine-dammed lakes aren't as important, and this section helps provide an explanation as to why we think moraine dams in Alaska are more stable than in other regions.

L188-189: How did the authors ensure that the rate of moraine-dam failures is free from reporting biases?

In our previous study where we looked at changes in lake area in Alaska over time (Rick et al., 2022), all moraine-dammed lakes that were lost disappeared due to infill, and we did not see evidence of moraine-dammed outburst floods. Additionally, speaking with experts in the region with observational experience, they have not witnessed moraine-dammed outburst floods.

L192-193: Is there any study that has systematically explored the nexus between valley shape and the abundance of moraines?

Unfortunately, we are not aware of any study that has systematically explored this relationship, but anecdotally from observing moraine structure in both Alaska and the Himalaya, this appears to be true. We believe it contributes more to moraine structure than necessarily abundance, and likely contributes to moraines being more stable (i.e. fewer moraine-dammed outburst floods) in Alaska.

L211: Does this mean that smaller lakes are more short-lived? Or that they are more difficult to detect? Please discuss possible causes for this observation.

We have removed/revised this section as we agree with the reviewer that the Post and Mayo data should not be used without validation.

Fig 4b: Is there any evidence of old strandlines in the empty lake basins? This could help to support the hypothesis of the rapid disappearance of lakes since the late 1960s (which I am still somewhat skeptical of).

We have revised this section to focus only on persistent lakes and manually derived lake areas, rather than relying on the Post and Mayo dataset to assess total lake numbers. However, we did find some really interesting examples of lakes that have been lost. It appears that vegetation growth has also been fairly quick such that basins that do not appear to have any evidence of a lake previously, did indeed hold a lake in the 1950s/60s. Here is particularly illustrative example:

[Figure redacted]

L228-229: Is this a subjective or objective definition of hazard? In other words, how did the return levels for these lakes change?

This was a classification provided by Post and Mayo (1971). Their definition of most hazardous was the largest lakes, with a look at downstream flood inundation.

L234: Be careful with this statement. The study by Jenson et al. (2022, The Cryosphere) showed that lakes can maintain high peak discharges even when lake volumes decline.

We recognize that the relationship between peak discharge and lake volume can be more complicated, though there is a general trend that decreasing volumes decrease potential peak discharge. We added text to address this nuance, acknowledging that the relationship is not always straightforward:

L263-267: “We also recognize that the relationship between lake volume and peak discharge can be more complicated. For example, the presence of ice within a lake can alter this relationship, decreasing storage capacity yet increasing pressure on the draining water^o. Considering these limitations, we use estimated peak discharge as a first order estimate of event magnitude to assess changes over time.”

L237: Please add a reference that systematically assessed this scaling relationship.

We have removed this sentence and focus instead on our estimated peak discharge.

L236-247: Why do the authors refrain from showing and discussing reported trends in peak discharge? Arguing with trends derived for a number of lakes would enrich this rather qualitative paragraph. I guess they are aware of this compilation? <https://www.weather.gov/aprfc/gdlMain>

Thank you for this suggestion. We have added analysis of estimated peak discharge, creating a regional relationship using the NWS dataset as suggested (Fig. S8). We also used this dataset to compare events captured by satellite imagery and events reported in the NWS dataset (Fig. S14).

L259-260: ‘critical next steps to fully characterize hazards from glacial lakes’: the title of this manuscript suggests that you already solved this problem? Again, I suggest to tone down the focus on hazard in this paper.

Thank you for this suggestion. We have toned down the focus on hazards, though we have significantly expanded our analysis of event magnitude.

L270-281 – Summary: largely repetition of the Abstract, and adds not much to the previous paragraphs.

Thank you for this observation– we have expanded the summary to include better context for why this study is important and how it can be applied to other regions globally.

L273: what is ‘dam integrity’?

We have changed the text to “dam structure”.

L278: ‘dramatically’: please remain subjective.

We have removed this phrase to remain more subjective.

L377: Why did the authors choose such a low threshold for cloud cover? In many satellite images along the coast of Alaska, I have seen that the ocean is covered by clouds while the mountains are cloud-free. Even with a threshold of 80-90%, parts of the image, and thus the lakes of interest, may still be cloud-free and allow measuring the frequency and approximating the timing of lake drainage. A higher threshold could be especially helpful in the early years of the study period, when only Landsat TM is available. In this context, the question arises why the authors did not use Landsat MSS imagery. These go back to 1972 and would make it possible to verify the lakes indicated in the Post and Mayo map.

As previously stated, we tried a variety of thresholds and chose 20% as a way to maximize the number of good images and minimize the number of unusable images in our dataset. Even at 20% cloud cover, 42% of the images had to be manually removed due to poor quality, and loosening the cloud cover threshold would significantly increase the number of poor images within the dataset. We decided to add aerial images from the 1950s (available through Earth Explorer) to expand a subset of our dataset, rather than rely on the questionable outlines from a digitized Post and Mayo map.

L385-392: Why did the authors refrain from mapping lakes areas? This information would add so much to the discussion of changing hazards. I also understand the term ‘drainage event’, but the authors could also evaluate downstream changes (erosion, aggradation) and river avulsions to determine if the drainage event was indeed rapid or catastrophic? Slow drainage would otherwise undermine the motivation that the hazard from ice-dammed lakes in Alaska is decreasing.

We mapped lake areas for three different time periods (where possible – limited by availability in the 1950s). Ideally, we would have mapped all “before” and “after” areas for each event, but for 1150 events, this is nearly impossible. Additionally, in some instances, the “after” image is not clear, and only a small portion of the lake is used to determine that the lake drained. Many of these lakes are heavily ice-filled, and when they drain it is nearly impossible to determine the minimum extent of the water. Figures S5, S11, and S15 show examples of lakes before and after a drainage and demonstrate the difficulty of outlining the minimum extent.

L416-421: Which likelihood function did the authors choose? The outcome ‘day’ needs to take values between 0 and 365 and demands a response other than the Gaussian.

For this analysis, we took the date full/empty dataset for each individual lake (Fig. S7), and randomly selected any DOY between when the lake was full and the lake was empty to be the date of drainage. We allow for all DOYs between these two dates (the colored bars in Fig. S7) to be equally as likely to be the drainage date as we do not know the precise day of drainage. In this way, we are able to allow for the largest uncertainty in trend in DOY over time. We use the 95% confidence levels of the linear models to determine whether or not we can identify a significant temporal trend through time.

Thank you again for your excellent suggestions that improved our manuscript, and for including references which were helpful while revising the manuscript. We appreciate the time and effort you've invested and the wealth of specific knowledge you have shared.

Dear Referee 2,

Thank you for taking the time to review our study and provide constructive feedback. Your suggestions have improved the quality and clarity of the paper. Below we provide detailed responses to each of your comments, with our responses in blue.

Reviewer #2 (Remarks to the Author):

The reviewed manuscript presents a retrospective analysis on the decreasing hazards from ice-dammed glacial lakes in Alaska over the past decades. This work was largely based on various remote sensing datasets and the GEE platform. The authors note a lack of systematic and spatially explicit observations of lake drainage events in Alaska and suggest an unbiased look at other cold regions, which does highlight the potential importance of such work. The results of this study would be of interest to the global GLOF research community and hazard policymakers. However, the manuscript lacks in-depth analysis and discussion on the drivers and impacts of ice-dammed GLOFs and the produced inventory could be further enhanced by adding more useful parameters. For the reasons above (as detailed below) I would recommend a moderate to major revision.

Thank you for providing a detailed and expert review of this manuscript. We have addressed all comments, with detailed responses below. In order to address the impacts of ice-dammed lake GLOFs, we have added additional analysis estimating lake volume and peak discharge based on lake area from three different time periods (1950s, 1980s, and 2010s). This allows for a systematic assessment of change over time, revealing that while the frequency of events has remained unchanged, there has been a decrease in the magnitude of events. This represents an important advance in our understanding of the hazards associated with ice-dammed lakes.

Major comments:

Introduction: GLOF drivers should be briefly mentioned in the introduction including mass movements into glacial lakes, extreme rainfall events, glacier surges (especially the case for ice-dammed lakes), seismic activity, and piping failures. Also, GLOF impacts on infrastructure and livelihoods could be more explicitly discussed (e.g., for downstream hydropower systems, extensive fluvial erosion and secondary landslides caused; see Li et al., 2022)

Thank you for these suggestions. We have added a mention of GLOF drivers in the introduction, which now reads:

“Glacial lake outburst floods (GLOFs) occur when a lake dam fails or is overtopped, releasing a large volume of water to downstream environments. These events can be triggered by mass movements into a lake (i.e. avalanche, landslide, rock fall), earthquakes, glacier calving, extreme rainfall or snowmelt, flotation of an ice dam, or degradation of an ice cored moraine.”

We also expanded upon GLOF impacts, which now reads:

“GLOFs can cause devastating impacts to downstream infrastructure (i.e. roads, bridges, hydropower systems), ecosystems (i.e. altering sediment flux, salmon habitat), and livelihoods (i.e. fishing, agriculture)³⁻⁷.”

Line 79 and the GLOF inventory: I checked the dataset and found that the inventory only includes dates (full/empty) and longitudes and latitudes of glacial lakes. It would be more helpful to include the information for the lake area

and also drivers and impacts of these lake drainage events. This could be challenging to detail all the 1150 events, but such information would indeed be more useful for the science community and policymakers.

Thank you for this insightful comment. We have added to the database information for each individual lake and the area, volume, and estimated peak discharge for the 1950s, 1980s, and 2010s.

Line 96-99: I feel that the simple correlation between detected GLOF events and imagery number does not sufficiently and explicitly demonstrate that increased image number is the sole driver of the increased reported GLOFs. The increased satellite images could be the key driver, but other factors like glacier surges and climate drivers should also be analyzed. An in-depth analysis of the GLOF drivers from a process perspective would enhance the argument.

Thank you for this suggestion. We performed an independent analysis using the bootstrap method to determine whether the trend over time is significant if the number of images per year were constant. This method finds a marginally decreasing trend over time (-0.006 ± 0.001 events per year), and implies that while there might be some physically based changes, those are small compared to the trend driven by an increase in imagery availability.

We investigated drivers of change (elevation, ice thinning rates, glacier thickness) and did not find any significant trends, apart from that all lakes are associated with thinning glaciers, though there is not a statistically significant relationship between thinning rate and rate of lake area change. While we agree that an important next step is to understand the drivers of the observed changes (i.e. earlier timing, smaller lakes), this analysis is outside the scope of this study.

Discussion: I find that glacier surge as a potentially important driver of ice-dammed lake outburst floods was not sufficiently discussed in the paper.

We appreciate your suggestion to further investigate the role of glacier surges as drivers of ice-dammed lake outburst floods in Alaska, particularly given the important role that they play in other regions, such as High Mountain Asia. However, from our assessment, this does not appear to be the case in Alaska. For this study, we manually reviewed time-series of satellite imagery of all ice-dammed lakes and did not observe a clear link between ice-dammed lake presence (or behavior) with glacier surging. Most ice-dammed lakes in this region form in deglaciating tributary valleys (rather than valleys blocked by a surging glacier) or along ice margins, such that whether or not a glacier surges has little impact on the formation of ice-dammed lakes in the region. We did not observe the formation of new ice-dammed lakes due to glacier surges within the period of study. Despite these overarching observations, we did look at the relationship between surging glaciers in Alaska and ice-dammed lakes using surge type glacier information from the Randolph Glacier Inventory. We found that only 6% of our ice-dammed lakes were associated with a known surging glacier. While surging classifications within the Randolph Glacier Inventory are limited both spatially and temporally, it is the most comprehensive resource that exists for the region. This finding supports the notion that surge behavior and ice-dammed lakes are not as closely tied in Alaska as they are in other parts of the world. This would be an interesting topic to revisit with an updated regional inventory of surging glaciers in Alaska.

Minor comments:

Title: maybe it should explicitly mention the ice-dammed glacial lakes - the focus of this study. Also, it should be 1980s or 1960s?

Thank you for this suggestion. We have changed the title to include ice-dammed lakes and be more explicit about the contents of our study.

Line 32: it is unclear what do you mean by the 'large infrastructure projects. hydropower projects or highways?

We have changed the text to be more explicit about what type of infrastructure we are referring to:

“GLOFs can cause devastating impacts to downstream infrastructure (i.e. roads, bridges, hydropower systems), ecosystems (i.e. altering sediment flux, salmon habitat), and livelihoods (i.e. fishing, agriculture)³⁻⁷.”

Line 46-47: similarly, outburst floods from the ice-dammed Lake Kyagar and Lake Merzbacher Lake in High Mountain Asia also occur more frequently, as compared to the moraine-dammed lakes (see Li et al., 2022)

Thank you for this suggestion, we have included references to Lake Kyagar and Lake Merzbacher.

Line 56: 330 drainage events?

Thank you for catching this, we have clarified to read “drainage events”.

Figs. 1-3: is it possible to also report the uncertainties associated with these frequencies and timing, areas, and volumes?

We have added the uncertainty for the trends in frequency over time, which supports the conclusion that there is no significant trend over time. The timing trend is determined using a 95% confidence interval (length of segment in Fig. S8).

Fig. 3: please add the date of the images and scale information in fig. 3a; the location of this Strandline Lake should be marked in Fig. 1 to facilitate the readers

Thank you for these helpful suggestions. We have added image dates and a scale bar to Fig. 3a. We have also added the location of Strandline Lake to Fig. 1.

Line 181-184: it seems not fair to compare with the moraine-dammed lakes in the Himalaya. Instead, it should compare with the ice-dammed lakes in the Karakoram (see Bazai et al., 2021)

Thank you for this observation. We have changed it so now ice-dammed lakes in AK are compared to ice-dammed lakes in HMA:

“The documented yearly average of ~33 events per year between 1985 and 2020 is 12 times greater than the regional rate of GLOFs from ice-dammed lakes documented in High Mountain Asia (average of 2.75 events per year between 1985 and 2020³⁸). This striking difference in frequency between Alaska and the Himalaya can at least partially be attributed to the abundance of ice-dammed lakes in Alaska with documented drainages (n = 106) as compared to the number of ice-dammed lakes in the Himalaya that have one or more documented drainage events (n = 21³⁹).”

Line 198-200: This could be true as the sediment supplies from retreating glaciers worldwide have been increasing since the 1950s (see Li et al., 2021)

Thank you for sharing this insightful reference to support the infill of lakes with sediment.

Line 228: please indicate the time period of the lake inventory reported by Post and Mayo. Did it cover the 1960s?

We have clarified in the text that that Post and Mayo dataset was created using imagery from the 1960s.

Lines 259-266: please consider adding a few sentences about the broad implications of this study (e.g. for future infrastructure development and cryospheric hazard mitigation; see also Emmer et al., 2022)

We have added some context for how our study can be used, including local river flood forecasting and decision making related to hazards, as well as providing a framework for the systematic analysis of other global regions.

L284-286: “This information is valuable for local agencies and decision makers to aid in making informed decisions pertaining to water resources and hazard mitigation.”

L300-302: “This study provides a framework for the systematic assessment of ice-dammed lake drainage events that can be applied to other regions globally.”

References mentioned:

Bazai, N. A., Cui, P., Carling, P. A., Wang, H., Hassan, J., Liu, D.,... Jin, W. (2021). Increasing glacial lake outburst flood hazard in response to surge glaciers in the Karakoram. *Earth-Science Reviews*, 212, 103432.

Emmer, A., Allen, S. K., Carey, M., Frey, H., Huggel, C., Korup, O., ... & Yde, J. C. (2022). Progress and challenges in glacial lake outburst flood research (2017–2021): a research community perspective. *Natural Hazards and Earth System Sciences*, 22, 3041–3061.

Li, D., Lu, X., Overeem, I., Walling, D. E., Syvitski, J., Kettner, A. J.,... Zhang, T. (2021). Exceptional increases in fluvial sediment fluxes in a warmer and wetter High Mountain Asia. *Science*, 374(6567), 599-603.

Li, D., Lu, X., Walling, D. E., Zhang, T., Steiner, J. F., Wasson, R. J.,... Bolch, T. (2022). High Mountain Asia hydropower systems threatened by climate-driven landscape instability. *Nature Geoscience*, 15, 520-530.

REVIEWER COMMENTS

Reviewer #1 (Remarks to the Author):

Dear editors, dear authors,

Thank you for giving me the opportunity to review the revised manuscript NCOMMS-22-30500A entitled "Decreasing regional hazards from ice-dammed lakes in Alaska since the 1950s" by Brianna Rick and co-authors. In reading the rebuttal letter and the revised manuscript, I found that the manuscript has been improved and many of my earlier comments have been carefully addressed through additional analysis of other data sources. The written and graphic presentation has also improved.

I found four major points that could deserve revision.

1) The authors have attempted to emphasise the decrease in GLOF hazard by assessing changes in GLOF size, including changes in lake area for the largest lakes that burst out, as well as lake volumes and peak discharges for all lakes at specific points in time. Despite efforts to estimate GLOF volumes and discharges, I can only encourage the authors to refrain from these estimates in their manuscript. Lake volumes can have orders of magnitude errors when estimated from lake areas using empirical power law relationships. When these errors are fully propagated in the estimated trends (rather than just providing a mean estimate of lake volume, as the authors have done here), changes in lake areas might have ambiguous trends, eventually becoming indistinguishable from zero, as shown (but hardly discussed) in the work of Shugar et al. (2020, see their Extended Data, Figure 1). Assuming that errors in lake volume are fully propagated for the argument, the peak discharges again have errors in orders of magnitude for a given lake volume, as the authors show very clearly in Fig. S9. Taking both uncertainties in lake volume and peak discharge into account, it is likely that some, if not all, of the derived trends in flood volume and peak discharge could be statistically indistinguishable from zero. In my opinion, the only indicator of the magnitude of GLOFs that can be reliably measured from remote sensing data is the change in lake area (although the authors do not provide an assessment of the uncertainty in lake area, which is fair to mention). The analysis of change in lake area makes sense as it uses a method similar to that of detecting GLOFs, and still supports the argument of changes in GLOF hazards. I therefore recommend the authors to focus more on changes in lake area. If they would like to include the analysis on changes in lake volumes and peak discharges, it is important to provide a detailed description of the model setup (which fitting algorithm was chosen? What are the errors in the model?) and use prediction intervals for their

point estimates and error propagation.

2) The authors argue that the availability of imagery is "the primary driver of the increase in detected drainage events" (L116-117). Considering their methodological design, this result is not very surprising, as the availability of imagery and the presence of clouds or ice on these images are the only parameters that could formally change the number of detected drainage events from a remote sensing perspective. However, what surprises me more is the fact that physical factors such as temperature changes (<https://akclimate.org/climate-change-in-alaska/>) or glacier thinning (e.g. Extended Data Fig. 4 in Hugonnet et al., 2021) (among many others) are not assessed and hardly discussed in the manuscript. I wonder if these data might show similar trends, possibly co-varying with the reported frequency of drainage events in Alaska. The example of Strandline Lake shows that the frequency of runoff events can indeed increase, and I am concerned that physical factors may play an underestimated role in the authors' assessment. I therefore recommend that a similar correction procedure be applied to other variables to see how much of the trend in drainage frequency can be explained without the effect of increasing image availability.

3) The procedure for correcting for possible under-reporting (i.e. the "criteria for adequate imagery"; L444-454) is still very difficult for me to follow, and I would appreciate it if the authors could explain this method in more detail. Intuitively, I would swap the criteria "least restrictive" and "most restrictive": The most restrictive case is when there is a *maximum* number of 18 days between the last cloud-free image before and the first cloud-free image after a drainage event and you can still detect the event. The least restrictive criterion is if you consider anything longer than 32 days and you can still detect it. In my opinion, the probability of recognising a drainage event in only 18 days is very low, because lakes could be veiled by clouds and ice. The more time that passes and the more observations (i.e. images) are available, the greater is the chance of detecting a GLOF, no? It would be good if the authors could clarify this question.

4) When assessing trends in GLOF timing, the authors need to change the distribution of the response variable (day of the year) to a distribution that maps the output on a circle. The day of the year is a continuous periodic response, e.g. located between $-\pi$ and π or in this case between 0 and 365. Note that calendar day 0 = day 365, and that the prediction of the mean of these two calendar days (which a Gaussian regression does) is not 182.5. You may find these links helpful:

<http://circstatinr.st-andrews.ac.uk/>,

<https://www.frontiersin.org/articles/10.3389/fpsyg.2018.02040/full>. I assume that the temporal

trends might change only slightly for GLOFs that occur in summer and have low temporal variance. The errors of using a Gaussian response might be higher for cases occurring in winter, sometimes at the end of one year, sometimes early in another year.

L41: 'event frequency': suggest to rephrase to 'frequency of reported events'.

L55: Lake No Lake is in British Columbia, not Alaska.

L62-64: Please add a reference to this statement.

L68: Is it plural? 'ice marginal lakes'?

L96-97: Did the authors predict peak discharges for all lakes or only those that had burst out in this period according to their remote-sensing analysis?

L116-117: As explained above, this phrase needs support from additional analysis to remain valid. In addition, the authors refrained from mentioning how many cases drained unnoticed in their method, but were reported by eyewitnesses or gaged by hydrological stations, as shown in Figure S14. This analysis could make a good contribution here.

Fig. 1: a - Please add the source of the basemap. c – "Launch" instead of "Lauch".

Fig. 2: a - Would be good to increase the size of the bubbles. b – Why did the authors stack the probability densities from individual lakes to a single probability density distributions? Much of the underlying variance gets lost with this representation. c – what is the bin size? Both in c and d, box or violin plots might be a more suitable choice, because they avoid the problem of binning the data into groups.

L164: 'this analysis' – which one?

L173-174: nicely rewritten, thanks.

Fig 3d: Is the estimated volume the volume before minus the volume after the GLOF? (in any case, I would refrain from this analysis as explained above).

L221: Suggest to remove 'Unfortunately'.

L237-238: Did you assess the thinning only for the part of the glacier that dams the lake? Would be good to trim the lake outline only to the part of glacier, given that local rates of thinning might be different to that of the entire glacier.

Fig. 4: a – please add the names of the lakes and add image credits. c – a number of lakes does not drain entirely. How do you account for partial drainage when estimating peak discharge? c – 'middle 50% of values' suggest to rephrase to 'encompass data between the 25th and 75th percentile' or something similar.

L253-255: Given the theory on changes in peak discharge in work of Jenson et al. (2022), as well as the reported discharges in the NWS database, I'm not sure whether I can fully support this statement.

L268-275: The authors could add in this paragraph that an earlier GLOF timing might change local hazard, because GLOFs might occur before seasonal high flows from spring melt?

L444-462: Please revise these paragraphs. The rationale and steps in the bootstrap method are entirely unclear to me.

L467: How did you stack the confidence intervals of the 10,000 simulations?

L477-478: Please add more information on these images, their resolution, whether they were already georeferenced, etc.

L483: Shugar et al. (2020) used two different equations, one for lakes smaller than 0.5 km² and one for lakes larger than 0.5 km². Why did the authors decide to use just one?

L488-496: This approach, if the authors would like to keep it in the manuscript, needs substantially more information on why a power law was used, which algorithms were used to fit the function, errors of the individual model parameters, etc.

Fig. S6: Please add IDs as used in the supplementary table on hydroshare.

Fig. S7: Please do not use red and green in the same figure.

Fig. S8: Why not place the panels one below the other and label them with their "real" IDs on the y-axis? This would avoid adding another table just for the names in this graph.

Fig. S10: Values seem to be a cut off along the x-axis.

Fig. S11: Again, please avoid red and green. I cannot differentiate these colors.

Fig. S12: Please add scale bars and image credits. The images seem to have higher resolution than Landsat.

Fig. S14: Why is the last panel in the third row empty?

Fig. S15: Please add scale bars and image credits.

Reviewer #2 (Remarks to the Author):

This is my second reading of the manuscript. The authors have appropriately addressed most of the earlier comments of each reviewer, and the result is a significantly improved paper. In particular, the revised work provides more useful GLOF data parameters including lake volume and peak discharge that should be helpful for the regional GLOF risk assessment and future risk mitigation.

As the authors also noted that there is a recent Nature study published, entitled "Less extreme and

earlier outbursts of ice-dammed lakes since 1900". This should somewhat unfortunately downplay the importance of this regional work. However, I would recommend publishing it, as it highlights that "A global compilation of ice-dammed lake drainage events reports 1569 events from 186 lakes between 1900 and 2020, whereas our analysis documents 1150 events from 106 lakes between 1985 and 2020 from just within the Alaska region. This suggests that many ice-dammed lake outburst floods have gone undetected globally, and warrants systematic study of other global regions." I would suggest that such comparison information should also appear in the abstract, as this would be a key contribution of this study.

In the conclusions, the authors highlight that they provide a framework for the systematic and unbiased assessment of ice-dammed lake drainage events. However, it is not entirely clear what is the framework? A quick search of the word 'framework' only appears twice in the main paper. Please the authors consider adding a flow chart detailing the framework in the SI and some more relevant methodology information in the main paper. This would be helpful for the subsequent works and increase your citations.

Further, the authors should acknowledge the following two recent papers regarding the GLOF community's perspectives and the impact of GLOF on erosion and sediment fluxes (and vice versa).

Emmer, A., Allen, S. K., Carey, M., Frey, H., Huggel, C., Korup, O., ... & Yde, J. C. (2022). Progress and challenges in glacial lake outburst flood research (2017–2021): a research community perspective. *Natural Hazards and Earth System Sciences*, 22(9), 3041-3061.

Zhang, T., Li, D., East, A.E. et al. Warming-driven erosion and sediment transport in cold regions. *Nat Rev Earth Environ* 3, 832–851 (2022).

Also, ref (23) and ref (41) are the same paper; please double-check the similar issue throughout.

Dear Reviewers,

Thank you for reviewing our manuscript “Decreasing regional hazards from ice-dammed lakes in Alaska since the 1950s”. We sincerely appreciate the time, effort, and thought that you have put into helping improve our manuscript. Please see our detailed responses below (in blue).

Reviewer #1 (Remarks to the Author):

Dear editors, dear authors,

Thank you for giving me the opportunity to review the revised manuscript NCOMMS-22-30500A entitled "Decreasing regional hazards from ice-dammed lakes in Alaska since the 1950s" by Brianna Rick and co-authors. In reading the rebuttal letter and the revised manuscript, I found that the manuscript has been improved and many of my earlier comments have been carefully addressed through additional analysis of other data sources. The written and graphic presentation has also improved.

I found four major points that could deserve revision.

1) The authors have attempted to emphasise the decrease in GLOF hazard by assessing changes in GLOF size, including changes in lake area for the largest lakes that burst out, as well as lake volumes and peak discharges for all lakes at specific points in time. Despite efforts to estimate GLOF volumes and discharges, I can only encourage the authors to refrain from these estimates in their manuscript. Lake volumes can have orders of magnitude errors when estimated from lake areas using empirical power law relationships. When these errors are fully propagated in the estimated trends (rather than just providing a mean estimate of lake volume, as the authors have done here), changes in lake areas might have ambiguous trends, eventually becoming indistinguishable from zero, as shown (but hardly discussed) in the work of Shugar et al. (2020, see their Extended Data, Figure 1). Assuming that errors in lake volume are fully propagated for the argument, the peak discharges again have errors in orders of magnitude for a given lake volume, as the authors show very clearly in Fig. S9. Taking both uncertainties in lake volume and peak discharge into account, it is likely that some, if not all, of the derived trends in flood volume and peak discharge could be statistically indistinguishable from zero. In my opinion, the only indicator of the magnitude of GLOFs that can be reliably measured from remote sensing data is the change in lake area (although the authors do not provide an assessment of the uncertainty in lake area, which is fair to mention). The analysis of change in lake area makes sense as it uses a method similar to that of detecting GLOFs, and still supports the argument of changes in GLOF hazards. I therefore recommend the authors to focus more on changes in lake area. If they would like to include the analysis on changes in lake volumes and peak discharges, it is important to provide a detailed description of the model setup (which fitting algorithm was chosen? What are the errors in the model?) and use prediction intervals for their point estimates and error propagation.

Thank you for bringing up this important point. We have now addressed uncertainty at each step. For each individual lake, we assume the error of the lake outline is +/- 0.5 pixel (a common metric used in other papers such as Fujita et al., 2009, Salerno et al., 2012, Zhang et al., 2015, Nie et al., 2017, Rounce et al., 2017, and Wang et al., 2020), and calculate this by multiplying the perimeter of the lake by half a pixel (15 m for the Landsat imagery in the 1980s and 2010s, and 1.5 m for the higher resolution historical

imagery from the 1950s). When assessing change in lake area over time, we propagate the uncertainty using the equation:

$$E_{diff} = \sqrt{E_x^2 + E_y^2}$$

Where E_x and E_y are the error for the first and second lake outlines, respectively. Lakes where the area difference was greater than E_{diff} were determined to have detectable area change. We found that 70% (85/121) of the lakes experienced significant change, and of those lakes, 79% decreased in area and 21% grew in area.

To account for uncertainty when calculating the volume of the lake, we used the Monte Carlo method for the lake area as well as to account for the uncertainty in the area to volume calculation. We do this by setting a range of values that the area could be (from area - error to area + error), and for each run we take a random number between those two values (using uniform distribution) to use as the area. We do a similar thing with the coefficients in the volume equation, setting a range of values (determined by the standard error reported by Shugar et al., 2020), and randomly selecting a value for the coefficients for each of the 10,000 runs. We then have a range of values for the volume of each lake, and use the 25th and 75th percentiles to define the most likely range in values for the volume of the lake. We then compare the range in volume values between the time periods, and if the two ranges do not overlap, we determine that there was a detectable change in volume (Fig. R1). Using this method, we find that of the 85 lakes with significant area change, 52 lakes (61%) have significant volume change.

Fig. R1. Change in calculated volume over time (Gray = 1950s, Red = 1980s, and Blue = 2010s).

We apply the same method to calculating the estimated peak discharge, selecting a random value for volume between the 25th and 75th percentiles calculated in the previous step. We then calculate the peak discharge using a range of values for the coefficients (again determined by the standard error). We do this

10,000 times and again take the range in values between the 25th and 75th percentiles. Comparing the range in peak discharge values between the 1980s and 2010s we find that 70% (60) of lakes had a significant change in peak discharge, with 73% (44) of those lakes decreasing in peak discharge.

Fig. R2. Change in calculated peak discharge over time (Gray = 1950s, Red = 1980s, and Blue = 2010s).

We added details to our methods about why we chose a power-law relationship (this is what has historically been used for decades, i.e., Jenson et al., 2022, Walder and Costa, 1996, Clague and Matthews, 1973) and included the standard error for each of the coefficients.

Now that we have fully propagated the uncertainty of the measurements, equations, and calculations, we can more confidently say that the majority of lakes have decreased in area, volume, and peak discharge between the 1980s and 2010s. This also makes sense logically, at minimum that if we see a decrease in area, we should see a decrease in lake volume.

2) The authors argue that the availability of imagery is "the primary driver of the increase in detected drainage events" (L116-117). Considering their methodological design, this result is not very surprising, as the availability of imagery and the presence of clouds or ice on these images are the only parameters that could formally change the number of detected drainage events from a remote sensing perspective. However, what surprises me more is the fact that physical factors such as temperature changes (<https://akclimate.org/climate-change-in-alaska/>) or glacier thinning (e.g. Extended Data Fig. 4 in Hugonnet et al., 2021) (among many others) are not assessed and hardly discussed in the manuscript. I wonder if these data might show similar trends, possibly co-varying with the reported frequency of drainage events in Alaska. The example of Strandline Lake shows that the frequency of runoff events can indeed increase, and I am concerned that physical factors may play an underestimated role in the authors' assessment. I therefore recommend that a similar correction procedure be applied to other variables to see how much of the trend in drainage frequency can be explained without the effect of increasing image availability.

We investigated temperature as a predictor of the number of events detected, and found that both the number of images and Alaska's mean annual air temperature (MAAT) have increased over time, co-varying with the frequency of events. We find a stronger relationship between number of images and events than MAAT and events, and when we calculate feature importance for predicting the number of events per year, the number of images has over 3x greater importance than MAAT. This suggests that there could be a relationship between temperature and event frequency (Veh et al., 2022 also report a weak linear relationship), though the number of images does a better job at describing the trend (77%) than MAAT alone (64%) or images and MAAT combined (73%).

Fig. R3. Calculated feature importance for predicting the number of events per year.

We also investigated additional characteristics of the ice dams from publicly available datasets within the Open Global Glacier Model (OGGM; Maussion et al., 2019) (i.e., ice velocity, glacier thinning, ice thickness) to see if there are any clear relationships between these variables and the number of events per lake or the change in lake area over time. We found a weak relationship between lake area change and ice velocity, but believe that a more thorough understanding of these relationships on a lake by lake basis would be well suited for a future study and is outside the scope of this study.

We have added text to address this point:

“We documented an increase in the number of *detected* drainage events over time (Fig. 1b). Coincident with this increase, there have been increases in satellite image availability, mean annual air temperature in the region, and an increased rate of glacier mass loss. We find a strong positive correlation ($R^2 = 0.95$; Fig. S1) between the number of lakes with adequate imagery available (see Methods) and the number of events detected. Mean annual air temperature also co-varies with imagery and frequency of events over time, though it was found to be a weaker predictor ($R^2 = 0.64$) of event frequency than imagery availability ($R^2 = 0.77$). We find no clear relationship between event frequency and lake area, ice thinning rate, or ice thickness. These findings suggest that increased image availability is the primary driver of the increase in number of detected events with time. Image availability increased markedly with the launches of Landsat 7, Landsat 8, and Sentinel-2 in 1999, 2013, and 2015, respectively (Figure 1c), with an average of 133 images per year between 1985 and 2000 and an average of 575 images per year between 2001 and 2020.”

Fig. R4. Diagonal correlation between variables from the Open Global Glacier Model (i.e., ice velocity = `itslive_v` or `millan_v`, ice thickness = `consensus_ice_thickness` or `millan_ice_thickness`, and ice thinning rate = `hugonnet_dhdt`) and our calculated lake areas (`Area_1984`, `Area_2016`), area difference (`Area_diff`), and number of events detected (`nEvents_PM`).

3) The procedure for correcting for possible under-reporting (i.e. the "criteria for adequate imagery"; L444-454) is still very difficult for me to follow, and I would appreciate it if the authors could explain this method in more detail. Intuitively, I would swap the criteria "least restrictive" and "most restrictive": The most restrictive case is when there is a *maximum* number of 18 days between the last cloud-free image before and the first cloud-free image after a drainage event and you can still detect the event. The least restrictive criterion is if you consider anything longer than 32 days and you can still detect it. In my opinion, the probability of recognising a drainage event in only 18 days is very low, because lakes could be veiled by clouds and ice. The more time that passes and the more observations (i.e. images) are available, the greater is the chance of detecting a GLOF, no? It would be good if the authors could clarify this question.

We apologize for the confusing terminology. This is the opposite of what the criteria is – least restrictive means that in one year there are two images that are *at least* 18 days apart, not that they have to be less than 18 days apart. We tried to demonstrate this in Figure S3, showing that the time between the first image and last image needs to be 18+ days (for the least restrictive), and if it is less than 18 days, it is not

considered detectable. Similarly, for the most restrictive criteria, the two images need to be *more than* 32 days apart, AND they need to occur around the time that the lake typically drains. Given the progression in these criteria, the least restrictive criteria resulted in the greatest number of lakes with adequate imagery per year while the most restrictive criteria resulted in the lowest number of lakes with sufficient imagery per year. We have attempted to clarify the text in our methods:

“We assessed the change in frequency of events over time using two independent methods. Method one looked at how many lakes had a documented drainage event versus lakes that had adequate imagery for each year, while method two explored whether a trend existed over time if image availability remained constant.

For method one, we used three different criteria of “adequate imagery” determined using the interquartile range (IQR), ranging from least to most restrictive, to assess whether there was a significant change in drainage frequency over time (Fig. S3). In our record, the distribution of days between images that captured a drainage event indicates an average of 28 days between images, and the empirical probability distribution indicates 18, 32, and 48 days for the 25%, 50%, and 75% quantiles, respectively (Fig. S2). The least restrictive criteria required that for any given year and any given lake, there had to be at minimum two images that were at least 18 days apart. The medium restrictive criteria required two images that were at least 32 days apart (> 50% chance of capturing an event; Fig. S7). The most restrictive criteria included the timing of the images, requiring two images that were at least 32 days apart and their dates bracketing ± 20 days of the median drainage day of year (calculated using the midpoint between date full and date drained for each event per lake). Given the progression in these criteria, the least restrictive criteria resulted in the greatest number of lakes with adequate imagery per year while the most restrictive criteria resulted in the lowest number of lakes with adequate imagery per year. The ratio of number of events detected to number of lakes with adequate imagery per year was used to normalize for image availability in each criteria (Fig. 1d).”

Fig. R5. Visual representation of the three different criteria used to define whether a lake had adequate imagery to detect an event in any given year. The three criteria includes the least restrictive criteria (two images that are at least 18 days apart), the medium restrictive criteria (two images that are 32 or more days apart), and the most restrictive (two images at least 32 days apart, taking the timing of the images into consideration).

4) When assessing trends in GLOF timing, the authors need to change the distribution of the response variable (day of the year) to a distribution that maps the output on a circle. The day of the year is a continuous periodic response, e.g. located between $-\pi$ and π or in this case between 0 and 365. Note that calendar day 0 = day 365, and that the prediction of the mean of these two calendar days (which a

Gaussian regression does) is not 182.5. You may find these links helpful: <http://circstatinr.st-andrews.ac.uk/>, <https://www.frontiersin.org/articles/10.3389/fpsyg.2018.02040/full>. I assume that the temporal trends might change only slightly for GLOFs that occur in summer and have low temporal variance. The errors of using a Gaussian response might be higher for cases occurring in winter, sometimes at the end of one year, sometimes early in another year.

Thank you for this suggestion. We have looked into circular statistics, and agree that this could be very important for events occurring in the winter. However, our window of timing for imagery is limited to the snow free period, May 30 to November 1, so we do not believe that circular statistics are necessary for this particular analysis as we are unable to capture events occurring in the winter due to snow cover and low imagery availability.

L41: 'event frequency': suggest to rephrase to 'frequency of reported events'.
We have rephrased this sentence. Thank you for the suggestion.

L55: Lake No Lake is in British Columbia, not Alaska.
Thank you for pointing this out- we have clarified the text to read: "Suicide Basin in Alaska, Lake No Lake in British Columbia, Lake Kyagar and Lake Merzbacher in High Mountain Asia)."

L62-64: Please add a reference to this statement.
We added a reference to Post and Mayo (1971), which describes several instances where a lake released in the winter and fractured river ice, creating ice jams that caused flooding.

L68: Is it plural? 'ice marginal lakes'?
We have changed the sentence to read: "In Alaska, moraine-dammed is the most common type of contemporary ice-marginal lake..."

L96-97: Did the authors predict peak discharges for all lakes or only those that had burst out in this period according to their remote-sensing analysis?
We made estimates for all lakes, as this is the "potential peak discharge" if the lake were to drain. There were 15 lakes that we estimated that we did not observe a drainage for.

L116-117: As explained above, this phrase needs support from additional analysis to remain valid. In addition, the authors refrained from mentioning how many cases drained unnoticed in their method, but were reported by eyewitnesses or gaged by hydrological stations, as shown in Figure S14. This analysis could make a good contribution here.

We believe our additional analysis looking at the feature importance for predicting the number of events supports that imagery availability plays a large role in the number of events we are able to detect. While we cannot rule out temperature as having an impact, we say that imagery is the "primary" driver. We have added a sentence to include MAAT discussion: "Mean annual air temperature also co-varies with imagery and frequency of events over time, though it was found to be a weaker predictor of event frequency than imagery availability."

We find that for the 20 lakes that overlap in observations, 282 events were previously documented, and our method captured 200 events. This means that the remote sensing method could miss up to 30% of drainages. This is a difficult comparison though because in some instances the satellite method misses events, and in some instances it captures many more than were previously documented. But in total, the remote sensing method had a false negative rate of 30%, supporting that the number of events we captured is a minimum. We have added text to this section looking at the missed events.

Fig. 1: a - Please add the source of the basemap. c – “Launch” instead of “Lauch”.

Thank you for catching this mistake! We have corrected the spelling of Launch and added that the basemap is from ESRI.

Fig. 2: a - Would be good to increase the size of the bubbles. b – Why did the authors stack the probability densities from individual lakes to a single probability density distributions? Much of the underlying variance gets lost with this representation. c – what is the bin size? Both in c and d, box or violin plots might be a more suitable choice, because they avoid the problem of binning the data into groups.

Thank you for this suggestion. We have increased the size of the bubbles in a. We chose to stack the probability densities due to a previous comment that we previously switched back and forth between regional analysis and individual lakes, and for consistency we show regional results in the main text, and keep the individual lake trends in the supplementary material (Figure S8).

L164: ‘this analysis’ – which one?

We have reworded this to read: “While we do not identify trends in event frequency when averaged across the study area, some individual lakes showed an increase in drainage event frequency with time”

L173-174: nicely rewritten, thanks.

Thank you!

Fig 3d: Is the estimated volume the volume before minus the volume after the GLOF? (in any case, I would refrain from this analysis as explained above).

Yes, for this example it is the volume before minus volume after the drainage event.

L221: Suggest to remove ‘Unfortunately’.

Thank you for this suggestion. We have removed “Unfortunately”.

L237-238: Did you assess the thinning only for the part of the glacier that dams the lake? Would be good to trim the lake outline only to the part of glacier, given that local rates of thinning might be different to that of the entire glacier.

Yes, we only assessed the thinning rate of the ice dam nearest to the lake, as this is the rate relevant for the ice-dammed lake and could be very different elsewhere on the glacier. We placed a point visually in the middle of the ice dam, and averaged values within a 100 m buffer, excluding values outside the glacier outline.

Fig. 4: a – please add the names of the lakes and add image credits. c – a number of lakes does not drain entirely. How do you account for partial drainage when estimating peak discharge? c – ‘middle 50% of values’ suggest to rephrase to ‘encompass data between the 25th and 75th percentile’ or something similar.

We have added the name of the lakes and image credits in the caption.

We state in our methods “Here we assume that the volume of water drained is equal to the estimated volume of the lake”, which is likely an overestimation of peak discharge. However, as a first order estimate and clearly stating our assumptions, we believe this is still appropriate.

Thank you for this suggestion – we have changed the text to the suggested wording.

L253-255: Given the theory on changes in peak discharge in work of Jenson et al. (2022), as well as the reported discharges in the NWS database, I’m not sure whether I can fully support this statement.

We believe that in general, this statement is still true. It is true that Jenson et al. (2022) show that this relationship might change if there is lake ice present, but we softened the language to say that a decrease in area “likely” results in a reduction in maximum peak discharge. Additionally, in their conclusions, Jenson et al. (2022) state: “*Despite complex relationships between glacier and basin hypsometry, remnant ice thickness, and discharge, we find nearly linear relationships between peak outburst flood discharge and total water volume for individual basins.*” Therefore, we believe it is reasonable to state that, in general, a decrease in total water volume (which can be implied from a decrease in surface area) results in a decrease in peak flood discharge.

L268-275: The authors could add in this paragraph that an earlier GLOF timing might change local hazard, because GLOFs might occur before seasonal high flows from spring melt?

Thank you for this suggestion. We have added some text that the timing of the drainage (i.e. before or after river ice breakup, during low or peak flows) can change the local hazards.

L444-462: Please revise these paragraphs. The rationale and steps in the bootstrap method are entirely unclear to me.

We have reworded these paragraphs in hopes of clarifying our methods and justification.

“For method two, we used a bootstrap method to determine whether we could identify a trend in event frequency if image availability was constant over time. We tested this by randomly selecting the same number of images each year (we tested both 30 images and 50 images per year) between 1985 and 2020, then calculating the number of events captured by those images, and fitting a linear trend in the number of events through time. We did this 10,000 times to capture the variability in slope, and reported the 95% confidence interval for the trend in frequency of events over time. We found that if the number of images per year is kept constant, there is no significant trend in frequency over time. This results in an average trend of -0.006 ± 0.001 events per year (Fig. S4), or -0.6 events per century, which is negligible in practical terms. These results support our finding of unchanged frequency of ice-dammed lake drainage events between 1985 and 2020, and that image availability is the biggest driver of the increase in documented events over time.”

L467: How did you stack the confidence intervals of the 10,000 simulations?

In this analysis, each simulation recorded the slope of the trend through time given the randomly selected days (between the before image and after image date), and then we took the 95% confidence interval of

that distribution of slopes to capture the range in slopes that could be possible given the variation in the data. We have added text to clarify our methods.

L477-478: Please add more information on these images, their resolution, whether they were already georeferenced, etc.

These images are aerial photographs, ranging in resolution from 0.3-1.5 m pixels. These images were manually digitized using up to 10 tie points each. We have added this information to the text.

L483: Shugar et al. (2020) used two different equations, one for lakes smaller than 0.5 km² and one for lakes larger than 0.5 km². Why did the authors decide to use just one?

We have altered our calculations to account for the two different equations depending on whether the lake is greater than or less than 0.5 km². To avoid any errors introduced by changing equations for a lake that grows or shrinks over this threshold, we determine which equation is used based on the lake area at the earliest time step.

L488-496: This approach, if the authors would like to keep it in the manuscript, needs substantially more information on why a power law was used, which algorithms were used to fit the function, errors of the individual model parameters, etc.

A power-law equation is what has historically been used, from Clague and Matthews (1973) to Walder and Costa (1995) to Jenson et al. (2022), and what we found to best fit the NWS data, which we used to fit the regional relationship. We fit a linear regression on the log transformed variables of flood volume and peak discharge. We added the standard error of the individual parameters to the methods text.

Fig. R6. Calculated relationship between peak discharge and flood volume for ice-dammed lakes in Alaska. Gray bounds indicate the confidence interval for the power-law equation. While the fit is not perfect, we believe this is a good first order estimate for peak discharge from flood volume.

Fig. S6: Please add IDs as used in the supplementary table on hydroshare.

Thank you for this suggestion. We added the LakeIDs in the caption of the figure.

Fig. S7: Please do not use red and green in the same figure.

We apologize for this oversight. We have changed the colors to avoid red and green.

Fig. S8: Why not place the panels one below the other and label them with their "real" IDs on the y-axis? This would avoid adding another table just for the names in this graph.

Thank you for this suggestion. We had originally removed the Lake IDs when this figure was in the main manuscript due to the clutter, but now that it is in the Supplement we have added the IDs back on the y-axis.

Fig. S10: Values seem to be a cut off along the x-axis.

Thank you for this observation. We have extended the axis to include all points.

Fig. S11: Again, please avoid red and green. I cannot differentiate these colors.

We apologize for this oversight. The colors have been changed.

Fig. S12: Please add scale bars and image credits. The images seem to have higher resolution than Landsat.

We have added scale bars. These images are from Sentinel-2 and we have added this information in the caption.

Fig. S14: Why is the last panel in the third row empty?

This was due to the lack of specific date for the events documented. We have changed this figure so that for any event that has only a year and no date, the event is placed at DOY 150.

Fig. S15: Please add scale bars and image credits.

We have added scale bars and noted that this is Sentinel-2 imagery.

Reviewer #2 (Remarks to the Author):

This is my second reading of the manuscript. The authors have appropriately addressed most of the earlier comments of each reviewer, and the result is a significantly improved paper. In particular, the revised work provides more useful GLOF data parameters including lake volume and peak discharge that should be helpful for the regional GLOF risk assessment and future risk mitigation.

As the authors also noted that there is a recent Nature study published, entitled “Less extreme and earlier outbursts of ice-dammed lakes since 1900”. This should somewhat unfortunately downplay the importance of this regional work. However, I would recommend publishing it, as it highlights that “A global compilation of ice-dammed lake drainage events reports 1569 events from 186 lakes between 1900 and 2020, whereas our analysis documents 1150 events from 106 lakes between 1985 and 2020 from just within the Alaska region. This suggests that many ice-dammed lake outburst floods have gone undetected globally, and warrants systematic study of other global regions.” I would suggest that such comparison information should also appear in the abstract, as this would be a key contribution of this study.

Thank you for this suggestion. We have included this information in the abstract and it now reads: “In just 35 years in one region, we document nearly 75% of the number of ice-dammed lake drainage events documented globally over 100 years. This suggests that outburst floods have historically been underreported and warrants systematic study of other global regions.”

In the conclusions, the authors highlight that they provide a framework for the systematic and unbiased assessment of ice-dammed lake drainage events. However, it is not entirely clear what is the framework? A quick search of the word ‘framework’ only appears twice in the main paper. Please the authors consider adding a flow chart detailing the framework in the SI and some more relevant methodology information in the main paper. This would be helpful for the subsequent works and increase your citations.

Thank you for this suggestion. We have added substantial text to our methods to further the reader’s understanding of what we did, and make our methods reproducible in another region. We believe the term

“framework” is still relevant here, providing at minimum a conceptual structure as to the why and how to investigate ice-dammed lake drainage events around the world.

Further, the authors should acknowledge the following two recent papers regarding the GLOF community’s perspectives and the impact of GLOF on erosion and sediment fluxes (and vice versa).

Thank you for these suggestions and we apologize for our oversight. We have added these references in appropriate locations within the introduction.

Emmer, A., Allen, S. K., Carey, M., Frey, H., Huggel, C., Korup, O., ... & Yde, J. C. (2022). Progress and challenges in glacial lake outburst flood research (2017–2021): a research community perspective. *Natural Hazards and Earth System Sciences*, 22(9), 3041-3061.

Zhang, T., Li, D., East, A.E. et al. Warming-driven erosion and sediment transport in cold regions. *Nat Rev Earth Environ* 3, 832–851 (2022).

Also, ref (23) and ref (41) are the same paper; please double-check the similar issue throughout.

Thank you for catching this mistake. These have been consolidated into one reference.

References

Clague, J. J. & Mathews, W. H. The Magnitude of Jökulhlaups. *Journal of Glaciology* **12**, 501–504 (1973).

Fujita, K., Sakai, A., Nuimura, T., Yamaguchi, S. & Sharma, R. R. Recent changes in Imja Glacial Lake and its damming moraine in the Nepal Himalaya revealed by in situ surveys and multi-temporal ASTER imagery. *Environmental Research Letters* **4**, (2009).

Jenson, A., Amundson, J. M., Kingslake, J. & Hood, E. Long-period variability in ice-dammed glacier outburst floods due to evolving catchment geometry. *The Cryosphere* **16**, 333–347 (2022).

Maussion, F., Butenko, A., Champollion, N., Dusch, M., Eis, J., Fourteau, K., Gregor, P., Jarosch, A. H., Landmann, J., Oesterle, F., Recinos, B., Rothenpieler, T., Vlug, A., Wild, C. T., and Marzeion, B.: The Open Global Glacier Model (OGGM) v1.1, *Geosci. Model Dev.*, 12, 909–931, <https://doi.org/10.5194/gmd-12-909-2019>, 2019.

Nie, Y. *et al.* A regional-scale assessment of Himalayan glacial lake changes using satellite observations from 1990 to 2015. *Remote Sensing of Environment* **189**, 1–13 (2017).

Post, A. & Mayo, L. R. Glacier Dammed Lakes and Outburst Floods in Alaska. *Hydrologic Investigations Atlas HA-455, USGS* (1971).

Rounce, D. R., Watson, C. S. & McKinney, D. C. Identification of Hazard and Risk for Glacial Lakes in the Nepal Himalaya Using Satellite Imagery from 2000 – 2015. *Remote Sensing* **9**, (2017).

Salerno, F. *et al.* Glacial lake distribution in the Mount Everest region: Uncertainty of measurement and conditions of formation. *Global and Planetary Change* **92–93**, 30–39 (2012).

Shugar, D. H. *et al.* Rapid worldwide growth of glacial lakes since 1990. *Nature Climate Change* **10**, 939–945 (2020).

Veh, G. *et al.* Trends, breaks, and biases in the frequency of reported glacier lake outburst floods. *Earth's Future* (2022) doi:10.1029/2021ef002426.

Walder, J. S. & Costa, J. E. Outburst floods from glacier-dammed lakes: The effect of mode of lake drainage on flood magnitude. *Earth Surface Processes and Landforms* **21**, 701–723 (1996).

Wang, X. *et al.* Glacial lake inventory of High Mountain Asia (1990–2018) derived from Landsat images. *Earth System Science Data Discussions* 1–23 (2020) doi:10.5194/essd-2019-212.

Zhang, G., Yao, T., Xie, H., Wang, W. & Yang, W. An inventory of glacial lakes in the Third Pole region and their changes in response to global warming. *Global and Planetary Change* **131**, 148–157 (2015).

REVIEWERS' COMMENTS

Reviewer #1 (Remarks to the Author):

Dear editors,

Dear authors,

Thank you for giving me the opportunity to comment on the manuscript NCOMMS-22-30500B by Brianna Rick and co-authors.

From my point of view, only a few minor to moderate issues could deserve revision. You can find those in the line-by-line comments below.

The only small but major issue that is still not adequately addressed after two revisions is the title of the study: the authors want to point out that the hazard in Alaska has decreased since the 1950s. However, they have NOT assessed hazard itself. Hazard is the probability (or a return period) of an event occurring. The authors have thoroughly assessed changes in drainage frequency and, to some extent, changes in the size of glacier lakes. However, they have not quantified the hazard as this would demand statements on how the probability of a given lake volume or discharge has changed with time. Clearly, changes in the magnitude and frequency of lake discharges also indicate changes in hazard, and the authors do a good job in discussing these changes (not least as they conclude: "Determining relationships for event magnitude and downstream impact are critical next steps to fully characterize hazards from glacial lakes, albeit challenging ones"). Yet any robust statement about changes in hazard would need a detailed assessment, probably from the field of extreme value statistics, to remain valid. Therefore, the title, as catchy as it is, needs revision. One solution would be to put the key findings in the title, e.g. "Unchanged discharge rate from smaller ice-dammed lakes in Alaska since the 1950s" or something. Of course, I leave the decision on the exact wording to the authors; I just cannot support the term "hazard" in order to remain consistent with the terminology in flood hydrology.

L19: Please clarify that this "third of global events" refers to previous inventories.

L23: What is this "one region"? The whole study region compared to the global catalogue? Or a sub-region (like St. Elias) compared to the regional catalogue?

L41: Change "present" to a specific year?

L77: Sub-region "Alaska"?

L104-107: "Ice-dammed lakes ... (see methods).": Repetition, can be deleted.

L113: Fig. S14: Unusual that one of the last items of the supplement is listed first in the main text.
Possibly re-arrange the order of the supplement as it appears in the main text?

L115-122: There is no information here or in the methods about where the data on annual air temperature, ice thinning rate or ice thickness came from, what type these data are, how they were extracted, and how the trends/ relationships were assessed. Please add this information in the Methods.

Fig. 1a: The colour scale of the bubbles is difficult to read. Why not just a single colour gradient, e.g. from white to dark blue?

L224-231: Earthquakes are often considered a possible trigger for GLOFs, and Alaska has had several large earthquakes in recent decades. Could they have an impact on lake outflows? For example, the Denali earthquake in 2002 did not produce a telltale spike in the drainage frequency graph. Perhaps add another sentence about the role of earthquakes in triggering GLOFs.

L248: "with an acceleration of mass loss towards the present": add exact period.

L261: "An historical": "A historical".

L261-268: please clarify that the definition of "most dangerous" lakes is based on a subjective assessment.

L446: perhaps add "subsets of more than" 14,000 satellites ...

L457-458: I am confused because the authors state that drainage events here can span more than a single season. This means that a lake can be full in one year and empty in another, but still be classified as a drained lake. However, in their response letter, the authors write, " our window of timing for imagery is limited to the snow free period, May 30 to November 1, so we do not believe that circular statistics are necessary for this particular analysis as we are unable to capture events occurring in the winter due to snow cover and low imagery availability." However, if the last cloud-free image of one year shows a full lake and the first available image of the next year shows an

empty lake, it is very likely that this drainage event occurred in winter. In the hydroshare data, I found several discharge events that covered the winter season; however, in Fig. S7 that shows the timing of lake drainages, I could not find any individual event that spanned the winter season. This issue needs to be clarified in the text, and if necessary, corrected in the trend statistics and figures.

L537-566: Using the IQR (i.e. only the inner 50% of the data) is a clever way to cut off the tails of the distribution and thus make distributions "significantly/credibly different" that would not be so if 95% of the distribution were assessed. This is nicely illustrated in the calculation of peak runoff, where the IQR is even used twice to severely constrain the values that the power law scaling relationship would predict. This constraint could be included in the challenges and limitations section, as the values at the end of the distribution in particular are of great importance in hazard assessment.

L583: Another limitation that should be added is the strict threshold for cloud cover. If the authors were to re-evaluate only the largest lakes in the record with a cloud cover of 100% (Fig. S12; which is what I did for 1 or 2 hours), they would find that all of these lakes had more outbursts during the study period. However, for the relevance of the manuscript, this might be only a minor issue, given that the authors stated that their number of drainage events is likely an underestimate.

These examples illustrate the complexity of determining GLOF hazards, as both event frequency and magnitude combine to determine the hazard.

Thank you for reviewing our manuscript. We sincerely appreciate the time, effort, and thought that you have put into helping improve our manuscript. Please see our detailed responses below (in blue).

Reviewer #1 (Remarks to the Author):

Dear editors,
Dear authors,

Thank you for giving me the opportunity to comment on the manuscript NCOMMS-22-30500B by Brianna Rick and co-authors.

From my point of view, only a few minor to moderate issues could deserve revision. You can find those in the line-by-line comments below.

The only small but major issue that is still not adequately addressed after two revisions is the title of the study: the authors want to point out that the hazard in Alaska has decreased since the 1950s. However, they have NOT assessed hazard itself. Hazard is the probability (or a return period) of an event occurring. The authors have thoroughly assessed changes in drainage frequency and, to some extent, changes in the size of glacier lakes. However, they have not quantified the hazard as this would demand statements on how the probability of a given lake volume or discharge has changed with time. Clearly, changes in the magnitude and frequency of lake discharges also indicate changes in hazard, and the authors do a good job in discussing these changes (not least as they conclude: “Determining relationships for event magnitude and downstream impact are critical next steps to fully characterize hazards from glacial lakes, albeit challenging ones”). Yet any robust statement about changes in hazard would need a detailed assessment, probably from the field of extreme value statistics, to remain valid. Therefore, the title, as catchy as it is, needs revision. One solution would be to put the key findings in the title, e.g. "Unchanged discharge rate from smaller ice-dammed lakes in Alaska since the 1950s" or something. Of course, I leave the decision on the exact wording to the authors; I just cannot support the term "hazard" in order to remain consistent with the terminology in flood hydrology.

Thank you for your valuable perspective. We have decided to change the title to better reflect the direct findings of our work. Our new title is “Unchanged frequency and decreasing magnitude of outbursts from ice-dammed lakes in Alaska”.

L19: Please clarify that this "third of global events" refers to previous inventories.

We have altered the text to read “...third of previously documented global events”.

L23: What is this "one region"? The whole study region compared to the global catalogue? Or a sub-region (like St. Elias) compared to the regional catalogue?

The one region is Alaska, and we have changed the text to reflect this: “In just 35 years in Alaska, we document...”

L41: Change "present" to a specific year?

Thank you for this suggestion. We have added the year to which the study was evaluated (2017).

L77: Sub-region "Alaska"?

This is a study (Wolfe et al., 2014) that only looked at one map sheet from Post and Mayo (1971), so it is a subset of ice-dammed lakes mapped in Alaska (538/750). We have clarified the text to read:

“...the disappearance of ~50% of a representative subset of Alaskan ice-dammed lakes between the 1960s and 2008.”

L104-107: "Ice-dammed lakes ... (see methods).": Repetition, can be deleted.

Thank you for this suggestion. We have deleted these sentences.

L113: Fig. S14: Unusual that one of the last items of the supplement is listed first in the main text. Possibly re-arrange the order of the supplement as it appears in the main text?

Thank you for catching this mistake. We have renumbered our supplemental figures and adjusted the text accordingly.

L115-122: There is no information here or in the methods about where the data on annual air temperature, ice thinning rate or ice thickness came from, what type these data are, how they were extracted, and how the trends/ relationships were assessed. Please add this information in the Methods.

We have added more information to the Methods:

“We also investigated additional characteristics of the ice dams within the Open Global Glacier Model (OGGM⁴⁵). Dam locations were manually chosen near the center of the ice damming each lake, and values were averaged within a 100 m buffer. We compared statewide mean annual air temperature⁴⁶, dam ice thickness⁴⁷, and dam thinning rate²⁵ to see if there are any clear relationships between these variables and the number of events per lake or the change in lake area over time.”

Fig. 1a: The colour scale of the bubbles is difficult to read. Why not just a single colour gradient, e.g. from white to dark blue?

Thank you for this suggestion. We have updated the color gradient from light to dark blue.

L224-231: Earthquakes are often considered a possible trigger for GLOFs, and Alaska has had several large earthquakes in recent decades. Could they have an impact on lake outflows? For example, the Denali earthquake in 2002 did not produce a telltale spike in the drainage frequency graph. Perhaps add another sentence about the role of earthquakes in triggering GLOFs.

We have added text that GLOFs can sometimes be triggered by earthquakes, and that earthquakes do occur in Alaska. However, recent literature states that earthquakes appear to be less impactful than has previously been thought:

From Emmer et al. (2022): “*While numerous GLOF susceptibility assessment studies consider earthquakes possible triggers of GLOFs, recent studies showed that very few GLOFs have actually been triggered by earthquakes globally (Kargel et al., 2016).*”

Emmer, A., Allen, S. K., Carey, M., et al.: Progress and challenges in glacial lake outburst flood research (2017–2021): a research community perspective, *Natural Hazards and Earth System Sciences*, 22, 3041–3061, <https://doi.org/10.5194/nhess-22-3041-2022>, 2022.

L248: "with an acceleration of mass loss towards the present": add exact period.

Thank you for this suggestion. We have added the exact time period, from 2000 to 2019.

L261: "An historical": "A historical".

We have corrected this error.

L261-268: please clarify that the definition of "most dangerous" lakes is based on a subjective assessment.

We added “*what they believed to be 32 of the most hazardous*”

L446: perhaps add "subsets of more than" 14,000 satellites ...

Thank you for this suggestion. We have added this text.

L457-458: I am confused because the authors state that drainage events here can span more than a single season. This means that a lake can be full in one year and empty in another, but still be classified as a drained lake. However, in their response letter, the authors write, " our window of timing for imagery is limited to the snow free period, May 30 to November 1, so we do not believe that circular statistics are necessary for this particular analysis as we are unable to capture events occurring in the winter due to snow cover and low imagery availability." However, if the last cloud-free image of one year shows a full lake and the first available image of the next year shows an empty lake, it is very likely that this drainage event occurred in winter. In the hydroshare data, I found several discharge events that covered the winter season; however, in Fig. S7 that shows the timing of lake drainages, I could not find any individual event that spanned the winter season.

This issue needs to be clarified in the text, and if necessary, corrected in the trend statistics and figures.

Thank you for pointing out this contradiction. After further consideration, we decided to exclude events in which the before/after images were in different years due to the large uncertainty of when the lake may have drained. It is true that a few lakes may have drained during the winter,

however, this is not commonly the case. For example, if a lake had a before image in July 2010 and an after image in July 2011, the midpoint between those two dates would be January 2011, though it is not likely that is when the lake actually drained. We found that including these types of events and allowing for winter drainages skewed the trends more negative (earlier events over time), which we believe falsely inflates the number of lakes that had a significant trend in timing over time. Therefore, we excluded events in the timing analysis where the images were not captured in the same year. We additionally limited the dataset to lakes which experienced 5 or more events to create more robust trends over time.

We have added text to our Methods section which reflects our new criteria.

L537-566: Using the IQR (i.e. only the inner 50% of the data) is a clever way to cut off the tails of the distribution and thus make distributions "significantly/credibly different" that would not be so if 95% of the distribution were assessed. This is nicely illustrated in the calculation of peak runoff, where the IQR is even used twice to severely constrain the values that the power law scaling relationship would predict. This constraint could be included in the challenges and limitations section, as the values at the end of the distribution in particular are of great importance in hazard assessment.

We have added an acknowledgement that using IQR ignores the possibility of extreme events: "For lake volume and peak discharge, we use the IQR to identify the most likely range of values for each lake, but acknowledge that values may fall outside this range. Therefore, the extreme ends of the distribution may not be captured and could impact hazard assessment."

L583: Another limitation that should be added is the strict threshold for cloud cover. If the authors were to re-evaluate only the largest lakes in the record with a cloud cover of 100% (Fig. S12; which is what I did for 1 or 2 hours), they would find that all of these lakes had more outbursts during the study period. However, for the relevance of the manuscript, this might be only a minor issue, given that the authors stated that their number of drainage events is likely an underestimate.

It is true that using no cloud cover threshold would enable us to likely document a few more events, however, the number of images added to be manually looked at would be massive. We have added text to the Challenges and Limitations section to acknowledge that we likely miss a few events due to this cloud cover threshold:

"Additionally, our use of a 20% cloud cover threshold likely excludes some viable images which may reveal a drainage event."

These examples illustrate the complexity of determining GLOF hazards, as both event frequency and magnitude combine to determine the hazard.